

# Characterization of trace gas emissions at an intermediate port

Aldona Wiacek[1,2], Li Li[1], Keane Tobin[1], Morgan Mitchell[1]

[1]Department of Environmental Science, Saint Mary's University, Halifax, Canada
[2]Department of Astronomy and Physics, Saint Mary's University, Halifax, Canada

## Abstract

Growing ship traffic in Atlantic Canada strengthens the local economy but also plays an important role in greenhouse gas and air pollutant emissions in our coastal environment. A mobile open-path Fourier transform infrared (OP-FTIR) spectrometer was set up in Halifax Harbour (Nova Scotia, Canada), an intermediate harbour integrated into the downtown core, to measure trace gas concentrations in the vicinity of marine vessels, in some cases with direct or near-direct marine combustion plume intercepts. This is the first application of the OP-FTIR measurement technique to real-time, spectroscopic measurements of $CO_2$, $CO$, $O_3$, $NO_2$, $NH_3$, $CH_3OH$, $HCHO$, $CH_4$ and $N_2O$ in the vicinity of harbour emissions originating from a variety of marine vessels, and the first measurement of shipping emissions in the ambient environment along the eastern seaboard of North America outside of the Gulf Coast. The spectrometer, its active mid-IR source and detector were located on shore while the passive retroreflector was on a nearby island, yielding a 455-m open path over the ocean (910 m two-way). Atmospheric absorption spectra were recorded during day, night, sunny, cloudy and substantially foggy or precipitating conditions, with a temporal resolution of 1 minute or better. A weather station was co-located with the retroreflector to aid in processing of absorption spectra and interpretation of results, while a webcam recorded images of the harbour once per minute. Trace gas concentrations were retrieved from spectra by the MALT non-linear least squares iterative fitting routine. During field measurements (7 days in Jul-Aug, 2016; 12 days in Jan, 2017) Automatic Identification System (AIS) information on nearby ship activity was collected manually from a commercial website and used to calculate emission rates of shipping combustion products ($CO_2$, $CO$, $NO_x$, HC, $SO_2$), which were then linked to measured concentration variations using ship position and wind information. During periods of low wind speed we observed extended (~9 hr) emission accumulations combined with near-complete $O_3$ titration, both in winter and in summer. Our



results compare well with a NAPS monitoring station ~1 km away, pointing to the extended spatial scale of this effect, commonly found in much larger European shipping channels.   We calculated total marine sector emissions in Halifax Harbour based on a complete AIS dataset of ship activity during the cruise ship season (May – Oct 2015) and the remainder of the year (Nov 2015 – Apr

2016) and found trace gas emissions (tonnes) to be on average 2.8% higher during the cruise ship season, when passenger ship emissions were found to contribute 18% of emitted $CO_2$, $CO$, $NO_x$, $SO_2$ and HC (0.5% off season).   Similarly calculated particulate emissions are 4.1% higher during the cruise ship season, when passenger ship emissions contribute 18% of emitted PM (0.5% off season).   Tugs were found to make the biggest contribution to harbour emissions of trace

gases in both cruise ship season (23% $NO_x$, 24% $SO_2$) and off season (26% of both $SO_2$ and $NO_x$), followed by container ships (25% $NO_x$ and $SO_2$ in off season, 21% $NO_x$ and $SO_2$ in cruise ship season), but then either cruise ships in third place in season or tankers in third place off season, both responsible for 18% of trace gas emissions.   While the concentrations of all regulated trace gases measured by OP-FTIR as well as the nearby in situ NAPS sensors were well below maximum

hourly permissible levels at all times during the 19 measurement days, we find that AIS-based shipping emissions of $NO_x$ over the course of one year are 4.2 times greater than those of a nearby 500 MW stationary source emitter and greater than or comparable to all vehicle $NO_x$ emissions in the city.   Our findings highlight the need to accurately represent emissions of the shipping and marine sectors at intermediate ports integrated into urban environments.   Emissions can be

represented as pseudo-stationary and/or pseudo-line sources.

## 1.0 Introduction

### 1.1 World shipping trends and emissions regulations

In 2015, seaborne trade is estimated to have accounted for more than 80% of total world

merchandise trade.   Seaborne trade volume expanded by 2.1% in 2015 (3.5% in 2014) and it continues to grow, albeit not without challenges (United Nations Conference on Trade and Development (UNCTAD), 2016, p. 25).   Shipping is also the most efficient transportation mode with the lowest Greenhouse Gas (GHG) emissions per ton of cargo and per km of transport as compared to rail, road and air (Seyler et al., 2017).   Between 2007 and 2012 shipping emissions

comprised on average only 2.8% of global $CO_2$-equivalent emissions (incorporating $CH_4$ and $N_2O$) from fossil fuel consumption and cement production (International Maritime Organization



(IMO), 2015); however, while other land-based sources work to reduce emissions, shipping emissions are projected to increase by between 50% and 250% by the year 2050, depending on economic and energy developments, as well as efficiency improvements.   The great majority (86%) of the above global emissions come from international shipping and are dominated by

container, bulk carrier and tanker emissions from main engine operations (as opposed to auxiliary engines and boilers).   Compared to $CO_2$-eq emissions, global $NO_x$ and $SO_x$ emissions from all shipping comprise a higher proportion of anthropogenic sources at 15% and 13% (IMO, 2015), respectively.   The majority of these emissions are again from international shipping and are projected to increase along with $CO_2$, with some exceptions.   $SO_x$ emissions will continue

to decline through 2050 because of the IMO International Convention for the Prevention of Pollution from Ships (MARPOL Annex VI) requirements on the Fuel Sulphur Content (FSC) in ships sailing globally and within Sulfur Emission Control Areas (SECAs, further explained below).   Emissions of $NO_x$ will be modulated by Tier I, II and III controls on ships sailing globally and within Nitrogen Emission Control Areas (NECAs, see below).   Notably, if fuels

shift to LNG, then methane emissions will increase rapidly (IMO, 2015).   The recent designation of the North and Baltic Seas as NECAs (World Maritime News, 2016) is predicted to lead to a greater shift to LNG in the shipping fleet fuel mix (Jonson et al., 2015).

   IMO regulations regarding $SO_x$ (and associated particulate matter, PM) emissions in a SECA historically required a reduction of FSC from the global average at the time of the SECA

entering into force to 1.5% (by mass) prior to July 2010, a further reduction to 1.0% after this date, and a final reduction to 0.1% after January 2015.   The actual historical FSC used in a given shipping area depends on when the relevant SECA began to be enforced.   The Baltic and North Seas SECAs began to be enforced in 2006 and 2007, respectively (Jonson, 2015), while the North American SECA (including most of Continental US & Canada) began to be enforced

in Aug 2012 (Environment Canada, 2012).   Thus in our study location in Halifax, Canada, the FSC of ships was reduced in Aug 2012 from an average value below the global FSC limit of 3.5% applicable at the time to 1.0%, and again in January 2015 from 1.0% to 0.1%.   The global FSC limit was recently approved to be reduced from 3.5% to 0.5% in 2020 (IMO Resolution MEPC.280(70)), adopted 28 October, 2016).

A similar complexity applies to IMO regulations regarding $NO_x$ emissions in NECAs.   The



North American NECA was fully implemented in August, 2012, with the implication that ships constructed after January 2016 have to comply with Tier III controls on $NO_x$ emissions for engine power greater than 130 kW.    Tier III controls are a function of engine speed (rpm) and lead to 80% lower $NO_x$ emissions as compared to Tier I controls (for ships built after the year

2000 but before 2011) and 75% lower $NO_x$ emissions as compared to Tier II controls (for ships built after 2011 but before 2016).    Outside of the North American and Caribbean NECAs Tier I and II controls apply globally, with Tier III controls coming into force in 2021 in the newly designated Baltic and North Seas NECAs.    Because of the long lifetime of ships, it will take approximately 30 years before full fleet renewal to Tier III compliance (Jonson, 2015).

### 1.2 Atmospheric chemistry, health and climate impacts of shipping

While shipping uses only 16% of the fuel used by the entire transportation sector it produces 9.2 times the $NO_x$ as the aviation sector and 80% of the $NO_x$ produced by vehicles (Eyring et al., 2005), primarily due to the high-temperature combustion of diesel engines combined with a lack

of strong regulation – until recently.    Where $NO_x$, CO and other hydrocarbon ship emissions occur in pristine marine boundary layers, a high ozone production efficiency results in increased background $O_3$ concentrations, increased OH and thereby reduced methane lifetime.    Up to 12 ppbv $O_3$ increases are calculated by Endresen et al. (2003) in a model study in summer in the North Atlantic and North Pacific.    In more $NO_x$-polluted regions the relative perturbation to $O_3$

concentrations is smaller but not negligible, i.e., 3-5 ppbv over Nova Scotia in July (Endresen et al., 2003, their Figure 10d).    The potentially large relative importance of shipping $NO_x$ emissions in intermediate ports and urban environments with relatively low background levels of $NO_x$ has been noted by Dalsoren et al. (2010), although in their work the effect is most apparent for the Atlantic Canadian port of St. John's, Newfoundland (their Figure 10), which has one

fourth the population of Halifax and likely even lower background $NO_x$ levels.    More recently, Aulinger et al. (2016) also found that shipping emissions significantly increased the incidence of daily 8-hr maximum $O_3$ exceedances in areas where they comprised a sizeable fraction of emitted $NO_x$.    At high $NO_x$ concentrations, $NO_x$ has a lifetime of hours to days against removal by $HNO_3$ formation and subsequent wet and dry deposition (Endresen et al., 2003) and via

nighttime loss pathways that involve first the formation of $NO_3$ then equilibrium with $N_2O_5$, followed by its hydrolysis to nitric acid (Vinken et al., 2011).



Shipping also leads to particulate emissions in the form of black carbon (BC) (Lack and Corbett, 2012), organic carbon (OC), hydrated sulfates and trace metals (e.g., Celo et al., 2015). These primary particulate emissions are elevated over shipping routes, while secondary aerosols, dominated by the inorganic fraction have a much further spatial reach (e.g., Aksoyoglu et al.,

2016).    The composition of the secondary inorganic fraction will shift from ammonium sulphate towards ammonium nitrate as shipping sulfur emissions decline, especially in regions with abundant ammonia emissions (Aulinger et al., 2016).

With 70% of shipping emissions occurring within 400 km of land (Corbett et al., 1999), cardiopulmonary and lung cancer mortality due to primary and secondary shipping emissions of

PM2.5 have been estimated at ~60,000 per year, concentrated in densely populated coastal regions as well as regions downwind of shipping lanes and ports (Corbett et al., 2007).    These mortality estimates will be reduced by legislated fuel sulfur content reductions and increased by the growth of the shipping sector and the continued emissions of primary particulates and $NO_x$ (leading to nitrogen-based secondary inorganic PM2.5 formation), as well as the formation of

other secondary aerosols.    A more recent study by Liu et al. (2016) based on higher Asian shipping emissions that are still largely unregulated, AIS-based activity data (see below), expanded causes of death (i.e., respiratory illness), and updated exposure-response functions has produced a similar range of East Asian PM2.5-related mortality (8,700-25,500 cases per year) as the earlier work of Corbett (1,000-32,000 cases per year in East Asia), with a further 5,800 –

12,000 mortalities due to secondary ozone.    In the busy North and Baltic Seas, Jonson et al. (2015) estimate 0.1-0.2 years of life lost (YOLLs) in areas close to major ship tracks at current emission levels while demonstrating the positive effects of recent and future regulations to curb $SO_x$ and $NO_x$ emissions.    Although shipping-related annual mortality estimates are low in Nova Scotia as a result of a low population density, the concentration of shipping-related PM2.5

estimated by Corbett et al. (2007, their Figure 1), which determines exposure and risk factors, is as high in Halifax as along other major global shipping routes, i.e., Northern Europe, the Mediterranean, and East Asia, further motivating the continued study of shipping emissions in our region as the regulations on $NO_x$, $SO_x$ and PM emissions evolve in a protracted international legal process.

The primary and secondary perturbations to tropospheric gases and aerosols lead to a small but highly uncertain climate forcing because of the offsetting warming effects from $CO_2$





emissions and secondary $O_3$ formation on one hand, and direct and indirect aerosol cooling effects and the cooling from $CH_4$ lifetime reduction on the other hand (Eyring et al., 2010). While greenhouse gas warming effects are global in nature due to their long lifetimes, direct and indirect aerosol cooling effects are regional (Liu et al., 2016) and expected to decrease as sulphur

emissions are reduced under increasingly restrictive legislation.

### 1.3 Shipping in Halifax, Canada

The Port of Halifax (Figure 1), Nova Scotia, was ranked 24[th] out of the top 50 NAFTA region container ports, based on container volume, and as such represents an intermediate port

environment. It is the fourth largest port in Canada (again measured by container volume) and one of the most important inbound port gateways in North America, connected to more than 150 countries via 18 direct shipping lines. According to the administering Halifax Port Authority (HPA, 2017), between 2012 and 2016 the total cargo (containerized or otherwise) handled at the port was 8.4 million metric tonnes per year, on average. In 2015/16 the port generated more

than 37,000 full-time equivalent jobs (>12,000 in direct port operations) which represents 8.3% of Nova Scotia's employed labour force, and is responsible for generating $3.6 billion in gross economic output ($1.9 billion from direct operations, $1.7 billion from exports). In addition to providing services for container, bulk, break-bulk, and ro-ro cargo ships, the Port of Halifax receives calls from more than 130 cruise ships carrying ~230,000 visitors to Halifax each year

(2012-2016 average). In 2017 cruise ship numbers rose sharply to 177, carrying up to 298,000 passengers (cruisehalifax.ca). This growth is a result of a broader strategy of continued investment in port facilities to support provincial growth targets for trade activity, tourism and aquaculture exports (HPA, 2017). Unlike many North American ports, the Port of Halifax is fully integrated into the city's urban core, increasing exposure and motivating the present field

study as well as future, longer-term measurements of trace gases in Halifax.

### 1.4 Previous studies of shipping emissions

Shipping emissions have been studied in laboratory test engines (e.g., Petzold et al., 2008; Reda et al., 2014), on-board from auxiliary engines operating at berth (Cooper 2003), on-board from main

engines operating at sea (e.g., Agrawal 2008a,b; Moldanova et al., 2009; Fu et al., 2013; Celo et al., 2015; Zhang et al., 2016), by intercepting plumes from aircraft (e.g., Sinha et al., 2003; Chen



et al. 2005; Lack et al., 2011; Berg et al., 2012; Aliabadi et al. 2016) and from other ships (e.g., Williams et al., 2009; Cappa et al., 2014) as well as by using combined sampling approaches (e.g., Petzold et al., 2008; Murphy et al., 2009). A large number of studies measured the effects of shipping emissions on air quality from some distance on land, either by in situ (e.g., Lu et al.,

2006; Marr et al., 2007; Poplawski et al., 2011; Alfoldy et al., 2013; Diesch et al., 2013; Pirjola, 2014) or remote sensing techniques (McLaren et al., 2012; Burgard and Bria, 2016; Merico et al., 2016; Seyler et al., 2017). Where gaseous emissions were reported, they were typically of NO, $NO_2$, $SO_2$, CO and total VOCs. All shipping emission studies relating to Canada (Lu et al., 2006, Poplawski et al., 2011, McLaren et al., 2012; Burgard and Bria 2016) focus on Canada's largest

port of Vancouver and cover time periods prior to 2010, when permitted FSC was at the globally applicable maximum of 4.5% (prior to January 2012) and only Tier I $NO_x$ emission controls were applicable to engines built after 2000 (the North American SECA and NECA were fully implemented only in August, 2012.) The contribution of shipping emissions to air quality in Halifax has been estimated previously by Hingston (2005) and Phinney et al. (2006). The former

was a bottom-up inventory approach while the latter was an analysis of $NO_x$ and $SO_2$ concentrations measured at the NAPS monitoring site (discussed in detail in Section 2.3) under winds prevailing from the marine geographic sector. They estimated $SO_2$ and $NO_x$ from shipping to contribute between <10% (Hingston 2005) and ~30% (Phinney et al., 2006) in Halifax under the high $SO_x$ emissions regime of 2002. Gibson et al. (2013) measured the mass and

chemical composition of PM2.5 in Halifax during a two-month field campaign in summer of 2011 and they report 47% of Halifax PM2.5 pollution as long-range transported (LRT), 27.9% as a mixture of LRT and aged marine aerosol, and 3.4% as shipping aerosol emissions. The total contribution of LRT-influenced PM2.5 pollution (75%) is broadly consistent with but higher than the work of Jeong et al. (2011), who estimate 56 – 65% of PM2.5 pollution as non-local based on

a nearly two-year time series of speciated PM2.5 measurements (April 2006 – January 2008). Both studies used PMF analysis of chemical markers supported by air mass back trajectory calculations and local wind analysis.

### 1.5 Aims of this study

The open-path FTIR remote sensing technique has typical detection limits of ~1-10 ppbv (dependent on path length and co-adding time), which is higher than in situ or UV-based remote



sensing techniques, however, all trace gases have infrared spectral signatures and many are suitable for quantitative analysis.   As such, the first aim of our study was to assess the signatures of individual or blended ship plumes from typical harbour activities in the urban core of Halifax, in concentrations of $CO_2$, $CO$, $O_3$, $NO$, $NO_2$, $NH_3$, $CH_3OH$, $HCHO$, $CH_4$, $N_2O$, $SO_2$, $HNO_3$, $HONO$,

$C2H6$, $C2H4$, $C2H2$ and other VOCs retrieved from infrared spectra recorded in close proximity to ship emissions.   Second, we compare our measurements with nearby (~1 km away) NAPS monitoring station measurements to better understand the effects of in situ vs. open-path sampling geometries, and the spatial extent of shipping emissions influence in Halifax Harbour.   Third, we estimate the contribution of ship vs. land-based sources to air pollution at an intermediate port with

relatively low background $NO_x$ concentrations (18 ppb annual average in 2015).   The fourth and final aim of our study was to establish a baseline of trace gas concentration measurements as NECA regulations begin to affect an increasing percentage of the fleet, and as shipping activity and shipping fuel mixtures evolve with time.   While baselines are recognized as important and necessary in future studies, currently Halifax air monitoring is limited to the NAPS program;

moreover, research in North America has focused on the West Coast and the Gulf Coast, making eastern seaboard measurements highly relevant.

## 2.0 Methods

### 2.1 Open-path FTIR measurements of trace gas concentrations

A wide variety of trace gas species can be measured with high temporal resolution during both day and night using the technique of open-path FTIR spectroscopy.   Measurements are also possible during significant fog and precipitation events due to the weaker scattering of infrared radiation by condensed phase fog and rain droplets in the beam.   The monostatic OP-FTIR configuration (co-located source and detector) has recently been applied to measure biomass burning emission

factors (e.g., Paton-Walsh et al.; 2014, Akagi et al., 2014), agricultural emissions (e.g., Flesch et al., 2016, 2017) and vehicle emissions (e.g., You et al., 2017).   Over the course of 7 days in July and August (2016) and 12 days in January (2017) a mobile OP-FTIR spectrometer was set up between Halifax Harbour and George's Island lighthouse to detect trace gas concentrations in the vicinity of marine vessel emissions, with the possibility of direct or near-direct plume intercepts.

To the best of our knowledge, this is the first application of this technique to trace gas measurements in a shipping environment.   Figure 2 shows the geometry of the OP-FTIR system



setup during the campaign in Halifax Harbour. The components comprise Bruker's "Open Path System" arranged in a monostatic configuration (Jarvis, 2003). The active broadband IR source is modulated by a low-resolution Fourier Transform Spectrometer (FTS) and passed to a modified transmitting 12" telescope, also serving as a receiving unit. Collimated radiation returning from

the retroreflecting cube cornerarray if focused on a broadband IR detector, in our system a Mercury Cadmium Telluride (MCT) element responsive between $700 - 6000$ cm$^{-1}$ with Stirling cycle cryocooling. A significant advantage of this configuration is that only the FTS-modulated returning radiation is detected, while unmodulated emitted atmospheric radiation in the wavelength range of the detector is ignored as DC signal by the signal processing electronics.

The separation between the telescope and retroreflector must be large enough that sufficient absorption for detection can be achieved, which is different for each target trace gas in accordance with its concentration and absorption cross section. In practice, one-way open paths greater than ~500 m lead to greatly diminishing signal returns due to imperfect beam collimation, which in our

system leads to overfilling the retroreflector array at and beyond separations of ~300 m. Moreover, with increasing atmospheric path, interfering absorption from water vapour and carbon dioxide increases and strongly overlaps target gas features. The 910-m optical path length (455 m physical separation) used in our study was dictated by the separation between the mainland and George's Island. The measurement path is well defined spatially in the planetary boundary layer

and bridges the spatial scales of in situ point measurements and newer space-based satellite measurements.

We used the maximum spectral resolution of our system (0.5 cm$^{-1}$), as is appropriate for sampling strongly Lorentz-broadened rotational-vibrational gas absorption features at 1 atm. To improve the Signal-to-Noise Ratio (SNR), we co-added 240 interferograms operating at 4 Hz to

produce a single spectrum once per minute. In an attempt to resolve finer plumes of ships moving directly within our line of sight we reduced the sampling interval to 10 seconds (40 co-added interferograms) in most of our summer measurements, which reduced the signal-to-noise ratio (SNR) by $1/\sqrt{6}$.

Spectral acquisition was carried out with Bruker's proprietary software while trace gas

retrievals were performed with a non-linear least squares (NLLS) iterative fitting routine (Griffith et al., 2012) using the MALT forward model (Griffith 1996) and MATLAB processing tools



developed in house. The NLLS retrieval derives trace gas concentrations from transmittance spectra through an iterative fitting process that minimizes a least squares cost function between measured and calculated spectra, taking into account target and interfering gas absorptions, spectrum continuum shape, and instrumental lineshape parameters describing line broadening and asymmetry under both ideal (finite OPD and FOV, apodization) and real (wavenumber shift, phase error, effective apodization) spectrometer conditions (Griffith 1996). The forward spectral model does not assume linearity in Beer's Law for absorbance vs. concentration, meaning that both weakly and strongly absorbing spectral features can be used in the analysis. Forward modeled spectra were calculated based on temperature- and pressure-dependent line-by-line absorption coefficients from the HITRAN 2012 database (Rothman et al., 2013) and real-time temperatures and pressures measured at the retroreflector end of the open path using a commercial weather station (Davis, Vantage Pro 2), which also recorded solar and UV radiation, wind speed and wind direction.

The inverse result is not unique because of noise in the spectra and represents the most probable set of trace gas concentrations, continuum coefficients and instrumental parameters given the measured spectrum. The full set of retrieval parameters used is shown in Table 1. Instrumental parameters of resolution and field-of-view were fixed but instrumental parameters of wavenumber shift, phase error and effective apodization were retrieved. Spectral continuum curvature was retrieved as a slope and intercept only but spectra were divided by the short-path background spectrum to remove complex continuum features that cannot be modeled by polynomial functions with good numerical stability. Retrieval parameters were optimized to result in minimal spectral fit residuals (RMS residuals typically ~0.5%), unbiased trace gas concentrations and variations in time that are uncorrelated with concentrations of other gases. Although FTIR retrievals are precise, the accuracy has been conservatively estimated as "well below 10%" by Smith et al. (2011) for species with strong absorption features ($CO_2$, $CH_4$, CO), mainly driven by the accuracy of spectral parameters, the MALT forward model, spectrometer alignment, pressure and temperature representative of the path average and retrieved parameter errors. For species with weak absorption features or subject to interference from water vapour these errors may be higher.

## 2.2 Weather effects in OP-FTIR measurements





While the OP-FTIR measurement tolerates considerable fog and precipitation, as discussed above, when IR signal levels drop to near zero then Root-Mean-Square (RMS) retrieval residuals increase and retrieved concentrations become very noisy. During August measurements (2016) the system experienced foggy and rainy conditions in the open path, which allowed us to determine a

threshold level of IR signal in the spectrum as an objective criterion to screen for heavy fog and rain. When IR signal levels dropped below 0.05 arbitrary units (a.u.) at 2100 cm$^{-1}$ from a more typical value of 0.4, retrieved $O_3$ concentrations became highly scattered (Figure 3). However, as long as IR intensities were above 0.05 a.u., retrieved $O_3$ concentrations varied but did not correlate with IR intensity, indicating a true sensitivity of the retrieval to atmospheric $O_3$ variations.

During the period of near-zero IR intensity the wind blew first from the south-east, then south, then south-west, i.e., not directly into the lightly shielded retroreflector (Figure 2) facing west. Therefore, while we did not have access to the lighthouse to visually inspect the retroreflector at the time, signal levels appear to have been primarily reduced on account of water droplets in the open path as opposed to coating the retroreflector cube corners, which can also happen under

strong winds towards the retroreflector.

## 2.3 NAPS measurements of trace gases

The National Air Pollution Surveillance Program (NAPS) was established in 1969 to monitor and assess the quality of ambient (outdoor) air in populated regions of Canada. The target air

pollutants include: CO, $O_3$, NO, $NO_2$, $SO_2$ and PM, reported hourly but available per minute from Nova Scotia Environment (NSE) upon request. VOCs are measured on a 6-day rotating cycle by 24-hour canister sampling and laboratory analysis. There are two NAPS stations in Halifax Regional Municipality: one in downtown Halifax ~300 m from Halifax Harbour, and another at a suburban background location ~ 11 km NE of Halifax Harbour. The downtown NAPS station

is ~ 1 km from the OP-FTIR measurement site (Figure 1) and thus measures very similar air masses, but subject to downtown vehicle traffic and the influence of flow around mixed-height buildings. Because of the shape of the Halifax Peninsula (Figure 1), ship emissions can most easily reach both the NAPS and OP-FTIR sites by north, north-east, east, and south-east winds, although the NAPS station is ~300 m (and three roads) inland from the harbour and as such is always sampling

a mixture of vehicle and marine emissions. South winds are likely to bring marine emissions to the OP-FTIR instrument from the South End Ship Terminal while bringing more direct downtown



vehicle emissions to the NAPS station. North-west winds are likely to bring a mixture of shipping and vehicle emissions to both OP-FTIR and NAPS measurement locations. Finally, west and south-west winds mostly likely bring vehicle emissions to both OP-FTIR and NAPS measurement locations.

The NAPS station is located on a relatively small (2-lane) but very busy downtown street that serves as a corridor for mainly light duty vehicles and 14 city bus routes, with a bus stop ~80 m away to the north and ~50 m away to the south. Gaseous air pollutants ($NO$, $NO_2$, $CO$, $O_3$, $SO_2$) were sampled through an inlet on the 4th floor of a building adjacent to the road, ~10 m above car exhaust and ~8 m above bus exhaust plumes. Therefore both hourly average and especially per

minute measurements of $CO$, $NO$, and $NO_2$ concentrations are strongly influenced by instantaneous traffic density. Measured $O_3$ concentrations are also expected to respond (inversely) to traffic density due to the fast titration by $NO$. However, the inverse response of $O_3$ is expected to be slower and more spread out in time than $CO$, $NO$ and $NO_2$ given the lifetime of $NO$ against the $O_3$ titration reaction is ~76 s at 298 K and 30 ppbv $O_3$ (4-6 minutes for conversion

of > 99% $NO$ to $NO_2$) [McLaren et al., 2012].

### 2.4 Emission plume detection

During field measurements, the information on ship positions was collected in real time from AIS signals transmitted by ships, as displayed by www.marinetraffic.com. The information included

ship type, deadweight, name, and tracks in and near Halifax Harbour (latitude, longitude, time, and speed, updated every 2 - 3 minutes when close to a land-based receiver). We used this information to calculate ship emission rates (kg/min) and correlate them in time with concentration variations measured by OP-FTIR in units of parts-per-million (ppmv) or parts-per-billion (ppbv) by volume, accounting for ship position, wind speed and direction. Specifically, we identified

ship emissions as containing enhanced $CO_2$, $CO$, $NO$, $NO_2$ and reduced $O_3$, as done by many other authors, e.g., Lu et al. (2006) in the detection of shipping plumes drifting more than 5 km inland in Vancouver.

### 2.5 Total emissions calculations

We used AIS information on ship type, cruising status, and deadweight to calculate exhaust gas emissions (kg/min) from different types of ships according to a commonly used parameterization



(United States Environmental Protection Agency (USEPA), 2000) for ship power and load fraction (Table 2) and energy-based emission factors (Table 3). A summary of vessel types in port, their deadweight, and the integrated emissions (tonnes) during the measurement period is shown in Table 4. $CO_2$ is the highest emitted gas by mass, with CO emissions comprising <0.5% of $CO_2$

emissions, as expected for high temperature combustion, which also leads to high $NO_x$ emissions. Calculated $NO_x$ emissions represent both NO and $NO_2$ expressed as $NO_2$ equivalent mass, assuming 100% conversion of NO to $NO_2$, for better comparison to emission inventories. Finally, $SO_2$ emissions were calculated for FSC of 0.1%, the maximum permitted for ships in port during the measurement period. As such, calculated $SO_2$ emissions represent a conservative estimate.

## 3.0 Results and Discussion

### 3.1 Overall characteristics of dataset

In 2016, open-path FTIR measurements of trace gases were conducted in summer conditions from July 12 - 15, and again from August 15 – 17. Winter observations (lower atmospheric water vapour and less spectral interference, reduced mixing layer height, slower photochemistry,

suppressed biogenic emissions) were conducted from January 23 - February 3, 2017. Figure 4 shows the distribution of ship activities during the three measurement periods based on AIS signals, while Table 4 shows a summary of ship types along with calculated total emissions (tonnes). Due to winter storms, the winter measurement was interrupted on January 24[th] to 26[th] and February 1[st]

to 2[nd], 2017. Retrieved concentrations of $CO_2$, CO, $NO_2$, $O_3$, $NH_3$, HCHO, $CH_3OH$, $CH_4$ and $N_2O$ are shown in Figure 5. All time stamps presented in this work are in UTC-4, that is, without daylight savings time (DST) in summer. Temporal resolution was 1 minute in winter and also in summer prior to 16:00 on July 13[th], at which point it was increased to 10 seconds. This caused reduced repeatability (increased scatter) and a bias in retrieved $NO_2$ values after 16:00 on July 13[th]

(not shown) but did not affect other gas time series ($CO_2$, CO, $O_3$, $CH_4$, $N_2O$) strongly on account of their higher information content (greater target absorption, less water interference) in underlying spectra. Finally, summer measurements were recorded at $15°C < T < 30°C$ and $60\% < RH < 99\%$ while winter measurements were recorded at $-10°C < T < 5°C$ and $44\% < RH < 97\%$.

Concentrations of $CO_2$ and all other retrieved gases except $O_3$ show (Figure 5) various degrees

of enhancement on July 13, August 16, January 30 and Feb 1 during broad periods (~9 hrs) of low wind speeds and suppressed mixing. $O_3$ is completely or near-completely titrated during these





extended time periods. CO shows the same broad enhancements but also many relatively narrow enhancements on summer afternoons, related to pleasure craft in the harbour (Section 3.5). $NH_3$ concentrations show some enhancement when $CO_2$ is enhanced but are highly variable in both summer and winter, as is HCHO, however, the latter experiences stronger relative enhancements

than $NH_3$, pointing to the proximity of more concentrated sources to the measurement location. $CH_3OH$ shows strong enhancements correlated with times of suppressed mixing, like HCHO, but only in winter, with summer background concentrations slightly higher than winter, especially on July 12 and 13. Finally, $CH_4$ and $N_2O$ are elevated when $CO_2$ and CO are elevated, implying similar sources.

We assessed the spectral signatures of several gases other than those reported in Figure 5. Because of strong interference from absorption by atmospheric water vapour, NO was impossible to retrieve reliably except in winter measurements during times of greatest enhancement (Jan 30 and Feb 2). As such, the time series of NO is not shown. Similarly, $SO_2$ is highly susceptible to strong water vapour spectral interference and not possible to retrieve reliably even with our long

path length of 910 meters – in part due to the low FSC (0.1% m/m) used by ships during our measurement period. We also retrieved $HNO_3$, HONO, $C_2H_6$, $C_2H_4$, and $C_2H_2$ with mixed success. $HNO_3$ (1235 – 1340 $cm^{-1}$) was subject to similar water interference issues as $SO_2$. HONO (1220 $cm^{-1}$ – 1300 $cm^{-1}$) was below detection limits at all times, even during the extended emissions accumulation periods on January 30, when water vapour interference was at a minimum.

$C_2H_6$ (2900 – 3005 $cm^{-1}$), $C_2H_4$ (940 – 960 $cm^{-1}$) and $C_2H_2$ (725 – 775 $cm^{-1}$) showed some accumulation during periods of low wind speed, however, the retrievals require further work and independent measurement verification, beyond the scope of this study.

### 3.2 July 13 accumulation of emissions

A strong enhancement of $CO_2$ was recorded from 00:00 through to noon on July 13[th], associated with light winds and an enhancement of CO, $NO_2$, $CH_4$ and $N_2O$, as well as a complete titration of $O_3$ lasting ~6 hrs (Figure 5d, 6a). $CH_3OH$ and $NH_3$ concentrations were enhanced in and around the time interval from 0:00 to 12:00 but do not correlate as strongly with the $CO_2$ enhancement as the other gases. HCHO was enhanced but only near the end of the broad $CO_2$

enhancement. The enhancement of $CO_2$ beginning after midnight and associated with $O_3$ titration is inconsistent temporally and chemically with plant respiration of $CO_2$, which would be





expected to occur earlier (from 18:00 onwards as is the case in data acquired with our system in a forest environment) and not affect $O_3$ concentrations. Winds were light and from the north to north-east, which is in the direction of built environments extending for ~10 km. The timing of the event is also not consistent with morning traffic emissions (on a Wednesday), which would be

expected to start accumulating after ~6:00, not earlier in the night.

    From ~0:00 to ~12:00 on July 13 a Bulk Carrier ship maneuvered north of George's Island at a distance of ~1.5 km to our measurement open path (Figure 6a). From ~0:15 to 7:00 a harbour service Oil Tanker navigated to the same area and refilled fuel to the Bulk Carrier (Figure 6b). At ~6:00 a Ro-Ro Cargo ship voyaged to 1.8 km north of the measurement open path (300 meters

north of the Bulk Carrier and Oil Tanker) and short-term cruised in that area until ~9:30 (Figure 6c). The tracks of those three ships were all arriving from the south or south-east towards the area north of George's Island and departing in the reverse direction (Figure 6).

    We calculated the theoretical emissions of trace gases (kg/minute) by each of the three ships in all operation modes (dockside, maneuvering, slow cruising and cruising) north of George's

Island in terms of $CO_2$, CO, $NO_x$, NO, $SO_2$, HC, and PM (Figure 7). While the ships vary considerably in gross tonnage and deadweight, they have comparable emission rates. As shown in Figure 6a, while the Bulk Carrier and Oil Tanker were north of George's Island and the wind was from the south there was no significant trace gas enhancement. Only when the light wind changed to north-east at 1:20 did the concentrations of CO and $NO_2$ increase, while $O_3$ decreased

to 0 ppbv due to titration by NO ($NO+O_3 \rightarrow NO_2+O_2$) in freshly emitted combustion plumes (e.g., Brown and Stutz, 2012). The concentrations of CO fluctuated in rough agreement with the wind direction and location of sources until 4:00 while $NO_2$ variations were smoother, possibly because of other chemical processes, e.g., once $O_3$ is completely titrated no further conversion of NO to $NO_2$ is possible, but nighttime conversion to $NO_3$, $N_2O_5$ and $HNO_3$ (McLaren et al., 2010) as well

as heterogeneous conversion to HONO and $HNO_3$ may be taking place (Wojtal et al., 2011). We note that on July 13th twilight occurred at 4:06 (UTC-4; 5:06 ADT) and sunrise at 4:41 (UTC-4; 5:51 ADT) at which time photolysis of $NO_2$ ($NO_2 +h\nu \rightarrow NO+O$) increased in importance (e.g., Jacob, 1999) and $O_3$ began increasing even though CO also increased from 4:00 – 5:00, as did the wind speeds. A Pilot and Military Patrol vessel passed under the open path at 4:15 and 4:19,

respectively, with only slight variations in CO. The increase of CO from 6:00 to 7:00, when the wind was north / north-east was caused by 1) the Ro-Ro Cargo arriving at 6:00, 2) the Oil Tanker





leaving at 6:35 and 3) traffic emissions increasing across the Harbour in Dartmouth.   By 7:30 the winds were very light again and the baseline concentration of CO was ~200 ppbv as compared to ~150 ppbv at 1:00, likely reflecting accumulating morning traffic emissions in both downtown Halifax and Dartmouth.

After 8:00 wind speed increased and wind direction changed slowly from north / north-east to east / south-east and finally south (Figure 8b, from time index 0 to 10).   A Pilot vessel passed under the open path at 8:09, but there was almost no effect on $CO_2$ and CO concentrations at that time (Figure 8a).   Trace gases over the measurement open path were no longer impacted by emissions from the Bulk Carrier and Ro-Ro Cargo (north of the open path) nor by Halifax /

Dartmouth vehicle traffic emissions, but instead first (8:20) by one slow cruising Navy Warship (Figure 8a) at ~0.7 km on nearest approach and two additional moored Oil Tankers at ~2.1 km to east / south-east (Figure 8a), and then (8:50) by six ships moored ~0.7 – 1.5 km to the south (Figure 8a), including the Oil Tanker that refilled the Bulk Carrier earlier, now dockside.   Winds from the south would also bring emissions from heavy duty diesel engines of trucks operating at the

container terminal in the south, and other loading and port vehicles and machinery.   We estimated the combined instantaneous emissions of $CO_2$, CO, $NO_x$, NO and $SO_2$ of the Navy Warship together with the two additional Oil Tankers based on available AIS status information and added them together as "East ships" in Figure 8c.   Some AIS information on Navy vessels is classified, so we assumed a deadweight of 5000 tonnes and designated its emission model as that of an

offshore supply ship (Table 2).   Similarly we estimated the combined instantaneous emissions of the six docked "South ships" (Figure 8c).   The CO emissions of the "East" and "South" ships are comparable, however measured CO perturbations at 8:20 with an east wind (Figure 8a), when the Navy ship slow cruised east of the measurement path with two additional oil tankers ~1.4 km further east, were larger and sharper than measured CO perturbations at 8:50 with a south wind

and six docked "South ships".   It appears that the least diluted emissions of the Navy Warship dominate the trace gas response at 8:20 also in terms of elevated $CO_2$, slightly elevated $NO_2$ and slightly decreased $O_3$.   On the other hand, trace gases detected in the open path at 8:50 show a more dilute response in $CO_2$ and CO but a more pronounced increase in $NO_2$ and decrease in $O_3$, consistent with the 1-5 minute transport time of "South ship" emissions to the measurement open

path.   We do not have information on any exhaust after treatment used by the vessels analyzed,



however, this would not affect $CO_2$ levels, only $NO_x$, CO, HC, $SO_2$ and particle levels (Pirjola et al., and references therein).

While our measurements of ship emissions showed HCHO was below detection limits from 2:00 to 6:00 when the Bulk Carrier and the Oil Tanker were maneuvering under winds favourable

for detection in the open path, it has previously been noted (Agrawal et al., 2008b; Reda et al., 2015) that HCHO is the dominant emitted aldehyde from ship engines burning both heavy fuel oil and distillates, and it has been detected in the field (Williams et al. 2009). The late morning HCHO enhancement (8:00 – 10:00) is positively correlated with CO and $NO_2$ and inversely correlated with $O_3$ from approximately 8:15 to 9:15 (peak HCHO ~5 ppbv), while winds remain

relatively light and from the south-east / south (Figure 6a, 8ab). At 9:15 HCHO, CO and $NO_2$ are markedly reduced while $O_3$ increases (as winds increase), after which point HCHO peaks again (~6 ppbv HCHO) at 9:30 together with CO and $NO_2$ while $O_3$ is slightly reduced. The HCHO is most likely of anthropogenic origin, with the majority likely from secondary production (Luecken et al., 2012) via oxidation of accumulated precursors ($CH_4$, other alkanes, alkenes and VOCs).

Our $CH_3OH$ measurements show an increase at 8:00 (Figure 6a) but no correlation to HCHO, and both gases are markedly reduced at 10:00 as wind speeds increase, bringing background marine air to the measurement path with background $O_3$ concentrations. Williams et al. (2009) postulated that a buildup of HCHO during the night may be important in leading to an increased source of $HO_x$ radicals in the morning, however, we do not find evidence of extensive HCHO accumulation

during the bulk of the July 13 $CO_2$ enhancement (from 2:00 to 7:00, Figure 6a) under direct ship influence of the measurement path and low wind speeds, but instead only from 8:00 to 10:00.

As briefly noted earlier, throughout the broad nighttime period of enhanced $CO_2$ concentrations (0:00 – 12:00) both $NH_3$ and $CH_3OH$ levels (detailed view in Figure 6a) were also enhanced but not clearly correlated with $CO_2$, except at ~2:00 (more likely due to ships) and again

from 6:00 to 7:00 (more likely due to vehicle traffic). It is difficult to attribute $NH_3$ enhancements to either ships or vehicles in the study area. In an older study Burgard et al. (2006) found $NH_3$ emissions of 0 g/kg (within error) in the exhaust gas of diesel engines operating on roads (~0.5 g/kg for gasoline engines), however, $NO_x$ reduction technologies for diesel engines are evolving, as are the exhaust emissions (Piumetti et al., 2015). While Suarez-Bertoa et al. (2014) measured

$NH_3$ emission factors from a single diesel car engine (equipped with Selective Catalytic Reduction (SCR), a Diesel Oxidation Catalyst (DOC) and a Diesel Particle Filter (DPF)) that were higher



than some gasoline car engines, Carslaw et al. (2013) show in a study involving 70,000 vehicles that $NH_3$ emissions are most important for catalyst equipped gasoline vehicles and SCR-equipped buses, with gasoline engine $NH_3$ emissions ~6 times higher than diesel engine $NH_3$ emissions for newer car models (>2010).   On the other hand, SCR in ships operating with HFO may also be a

source of ammonia in the exhaust due to ammonia slip, currently not regulated in ship applications by the IMO (Lehtoranta et al., 2015).

Finally, $CH_3OH$ has strong biogenic sources related to plant growth and decaying plant matter, as well as from the marine biosphere, which is a large gross source but an overall net $CH_3OH$ sink (Hu et al., 2011).   It is also formed from CH4 oxidation, which is also a globally important source

of CO and HCHO, however this reaction is relatively slow and proceeds mainly in low $NO_x$ environments (de Gouw et al., 2005).   $CH_3OH$ has strong primary and weak secondary urban sources (de Gouw et al., 2005) and the main sink is by OH oxidation (Hu et al., 2011).   Rantala et al. (2016) show $CH_3OH$ concentrations correlated with traffic emissions in all seasons and of 100% anthropogenic origin during the winter and $42 \pm 8\%$ during summer, when biogenic

influences play a large role.   Our measured $CH_3OH$ concentration shows the same nearly monotonic rise from 6:00 to 7:00 as $NH_3$ and CO, when light winds from the north and north-west brought traffic emissions to the measurement path.   Throughout the July 13 enhancement of $CO_2$, $CH_3OH$ was elevated but highly variable, showing only occasional correlations with narrow CO spikes (Figure 6a) and its enhancement extends both before 0:00 and after 12:00 on July 13 (Figure

5g).   This suggests a blending of biogenic, vehicle traffic and shipping emission sources in the summer data.

### 3.3 January 30 accumulation of emissions

A strong enhancement of $CO_2$ was also recorded on January 30 from 5:00 to 14:00 (Figure 5) on

a windless morning (Figure 9) with busy harbour activities of 23 vessels (Figure 10 and inset). Ship and other emissions were accumulated in the surrounding area and were measured by the OP-FTIR system (Figure 9) as well as the NAPS monitoring station ~1 km away (Figure 11, Section 3.4).   There were no ships maneuvering or moored immediately north of George's Island on January 30 like on July 13 but instead vessels cruised in the shipping channel (Figure

10), including through the measurement open path.   From 5:00 to 9:00 concentrations of $CO_2$ and CO rose nearly monotonically as Monday rush hour mounted, with a perturbation between





6:00 and 6:30 from Container ship 1 as it navigated to the North End Terminal (5.9 km to OP-FTIR) passing outside of George's Island at 6:00. During this half hour period $NO_2$ increased and remained elevated until 10:00 while $O_3$ reduced to 0 ppbv and remained titrated until 9:30. Sunrise occurred at 7:35 (twilight at 7:03). The variable but increasing $NH_3$ concentration from

6:00 to 9:00 indicates that those air pollutants might be from the accumulation of emissions from vehicle traffic due to the mounting rush hour. There is also a very distinct accumulation of $CH_3OH$ during this time, which has known correlations with vehicle traffic and may be related to gasoline methanol content (Rantala et al., 2016) but also to windshield wiper fluid being used more frequently in winter months (Carrière et al., 2000), while accumulation is less apparent for

HCHO.

From the time series profile of CO at the time of Container ship 1 passing by our measurement location (heading N-NW) we infer that a weak breeze from the north-west must in fact have been present, even though the wind sensor is registering 0 km/h, with very occasional readings of 1.6 km/h and 3.2 km/h (Figure 9). This made it possible for the open path to sample

emissions from the north-west, which was the heading of Container ship 1, which changed status to maneuvering at 6:34, moored at 6:54 and became dockside at 7:19. At this point in time the vessel was located in the North End Terminal, ~6 km away, and separated from the open path by a portion of the land mass of Halifax peninsula (Figure 1), which would make the attribution of concentration changes solely to shipping emissions implausible. However, emissions from

ships can rise 2-10 times the stack height and experience vertical mixing on time scales of 20-40 minutes, possibly filling the entire boundary layer height, depending on buoyancy flux and boundary layer stability (Chosson et al., 2008). As such, it is not impossible to be detecting some shipping emissions from several km away on a still January morning.

A Pilot ship also crossed the measurement path at 6:15, and again at 9:41 and 11:01 (Figure

10) but did not produce significant signatures in trace gas concentrations. Also in the early morning a Navy Coastal Defense Vessel (55.3 m length) changed status from mooring to maneuvering at 6:28 at its berth location 2.1 km north-west to OP-FTIR (Figure 10). It left the berth in slow cruising mode at 8:31 until 9:03 when it reached the Bedford Basin 8.6 km north-west to OP-FTIR. Based on the very low wind speed, it would take 1-2 hours for its departure

emissions to advect to the measurement open path. The Navy Defense vessel returned from the Bedford Basin at 11:33 in slow cruising mode, arriving at berth at 12:15 in a slightly closer (1.9





km to OP-FTIR) location and turning off its engines at 13:00.

From 9:50 to 10:10 concentrations of HCHO were significantly elevated, with a modest increase in CO, a decrease in $NO_2$ and an increase in $O_3$ during the same time.   A Navy Warship (134.2 m length) started maneuvering near its berth (2.3 km to OP-FTIR) from 9:41 to

10:03 and then left the berth (Figure 10) in slow cruising mode at 10:10 heading for the Bedford Basin (8.9 km to OP-FTIR).   Under the light winds it would take ~1 hr for emissions from the Warship to reach the measurement open path, experiencing dilution along the way, thus making it unlikely that the sharp rise in HCHO at 9:50 (Figure 9) is due to the Warship.   As already mentioned, a Pilot ship also crossed the measurement path at 9:41 heading south east, however,

the same Pilot crossed the open path at 6:15 and 11:01 (Figure 10) with a similar speed and registered no enhancement in HCHO at those times (Figure 9).   As such, the origin of the strong HCHO enhancement is ambiguous and requires further field study.   An event of increased $CO_2$, CO, $NO_2$ and $CH_3OH$ at 11:50, with a more broad increase in HCHO over the next 20 minutes is also not clearly related to ship activities, motivating further field study.

From 11:00 onwards increasing wind speeds and mixing dissipated accumulated pollutants, while the wind direction changed to eastern winds after 12:00 (Figure 9).   At 7:00 Container ship 2 voyaged from the mouth of the harbour to the South End Terminal (1.2 km to OP-FTIR), however, the emissions of this vessel are unlikely to impact the measurement path due to the weak north-west breeze, as are the emissions of a General Cargo ship at 11:00 (3 km to OP-

FTIR).

### 3.4 Comparison to NAPS in situ data in downtown Halifax

Our OP-FTIR measurements were conducted ~1 km from the downtown Halifax NAPS station influenced by vehicle traffic, as described in Section 2.3 and Figure 1.   In summary, the NAPS

site is influenced by vehicle traffic emissions during all wind conditions, including those most directly from the harbour (N, NE, E, SE, S), which lies ~300 m to the east and three busy streets away from the NAPS station.   In contrast the OP-FTIR system is most directly influenced by ship emissions, except under winds from the downtown core (NW, W, SW).   The OP-FTIR measures trace gas concentrations in a 455-m path average (in this particular measurement campaign) and is

thus less sensitive to instantaneous emissions, which are unlikely to fill the entire open path. Figure 11 shows a comparison between in situ NAPS and OP-FTIR $NO_2$, CO and $O_3$



concentrations from January 23 to February 3, 2017. Similar variations in trace gases are apparent, e.g., the long-duration $NO_2$ and CO enhancements with a simultaneous depletion in $O_3$ on January 30 and February 1. This points to the extended spatial scale of this effect, which has been described as very local or near-field in land-based studies using in situ detection methods

(Merico et al., 2016; Diesch et al. 2013; Eckhardt et al., 2013).

There are, however, some notable differences between open path and in situ measurements. First, NAPS in situ measurements show multiple $NO_2$ values in excess of 50 ppbv, whereas the OP-FTIR system measurements do not, likely due to a combination of 1) strong vehicle emissions present at the NAPS site but not at the OP-FTIR site, 2) path-averaging of direct emissions by the

OP-FTIR system, 3) chemical transformation of $NO_2$ and 4) advection and dispersion of emissions between the measurement sites. Second, NAPS in situ measurements also show strong enhancements in CO values in excess of 500 ppbv, whereas the OP-FTIR measurements do not, however, the baseline of OP-FTIR CO measurements is higher than NAPS by 50-100%, with greatest differences during early morning periods before rush hours start. This large bias is

outside of the range of errors of either technique, estimated conservatively at ~10% for OP-FTIR and 15% for NAPS in situ measurements as per the *minimum* NAPS data quality objectives for CO (Canadian Council of Ministers of the Environment, 2011), however, the particular sensor accuracy at the NAPS site in question is <2%. If we take CO as a marker of fresh combustion emissions then persistently elevated CO at the OP-FTIR site should correspond to persistently

lower $O_3$ concentrations but in fact the reverse is observed (Figure 11), with OP-FTIR $O_3$ typically 35% higher than NAPS, except during periods of complete $O_3$ titration on January 30 and February 1. It is not clear what combination of emission differences, in situ vs. path-average sampling differences, chemical transformation as well as advection and/or dispersion is causing the persistent CO and $O_3$ biases, which are also evident in summer measurements (not shown).

While beyond the scope of this paper, the representativeness of in situ vs. path-average surface measurements is under further investigation and has also been documented by You et al., (2017).

### 3.5 Emissions from small craft compared to large ships

In the summer time series of CO concentrations, especially in late afternoons and early evenings,

there are many relatively narrow CO spikes (~1-15 minutes, > 250 ppb, Figure 5b) that do not match AIS-based ship activity. We noted pleasure craft in the harbor during the campaign but



were able to obtain per minute screenshots from a publically viewable webcam installed on the roof of a nearby tall building (http://www.novascotiawebcams.com/en/webcams/pier-21/), which included the measurement open path.    The images confirmed that many CO enhancements were caused by high speed pleasure craft when they passed under the open path – despite the path-

average nature of the OP-FTIR measurement and the short, 10-second acquisition time on the afternoon of July 13, 14 and August 15, 16.    Yet this is consistent with higher CO and lower NOx emissions of gasoline engines (predominant in small craft) as opposed to diesel engines (Henry, 2013).    On July 13 sea breeze winds were steady from approximately the south-east and relatively constant at ~20 km/h.    Figure 12 shows the effects of small craft and large ships in measured

concentrations of CO, $NO_2$, and $O_3$ as well as in images from the public webcam.    With south-east 20 km/h winds trace gas emissions from the 1-3 km visible area to the south-east of George's Island would take ~3-9 minutes to be transported towards the open path.    As such we found that three out of four significant $NO_2$ enhancements and correlated $O_3$ depletions lasting ~15 minutes were associated with large ships 1-3 km to the south-east, except at 20:00 (Figure 12).    Speed

boats crossing the open path had the most effect on CO and the least effect on $NO_2$ and $O_3$, as expected, given no time for NO titration of $O_3$ to proceed.    One small speed boat had a very pronounced effect on CO (Figure 12, 17:06) that lasted ~1.5 minutes and is consistent with the minimum time to traverse the measurement path at 20 km/h (1.4 minutes).    In winter, there were no high speed pleasure craft in Halifax Harbour and there are correspondingly fewer relatively

narrow CO spikes despite the longer time series (Figure 5b).    Our detection of a variety of CO enhancement signatures is in contrast to Diesch et al., (2013) who report none using an in situ technique with an 80 ppb detection limit during a study on ship traffic on the Elbe.

### 3.6 Distribution of emissions in space, time and by ship type

We calculated ships emissions in Halifax Harbour during the OP-FTIR measurement periods (Table 4) using the emission model from Tables 2 and 3.    Figure 13 shows the spatial distribution of $NO_x$ emitted during July and January measurements, with emissions clustering predictably around the North End, Richmond and South End Terminals, Navy Dockyards, Waterfront Wharves, harbour anchorage areas, as well as two Oil Terminals and an Autoport on the Dartmouth side of

the harbour.    Emissions of $NO_x$ from the South End Terminal are relatively lower in winter due to a lack of cruise ship activity, while emissions from one of the two Oil Terminals in Dartmouth



are relatively higher in winter, likely because heating with fuel oil is common in the city.   (Tanker emissions account for ~15% of $SO_2$ and $NO_x$ emissions in "summer" and ~19% of $SO_2$ and $NO_x$ emissions in "winter", as shown in Table 5 and further discussed immediately below).   $SO_2$ emissions follow a very similar pattern to $NO_x$ emissions and are not shown.   As noted, municipal

passenger ferry and pleasure craft emissions are not included in the AIS data used in our study, therefore, where emissions appear near ferry terminals they are caused by other vessels such as tugs and coastal supply ships that use the same terminals.

To account more fully for ship emissions by season and type we calculated emissions for a full year using AIS activity data from May 2015 to April 2016, divided into a broad "summer"

season when cruise ships are active (May 2015 – October 2015) and a broad "winter" season when they are not (November 2015 – April 2016).   The distribution of $NO_x$ and $SO_2$ emissions (Figure S1) is very similar to the short measurement time periods shown in Figure 13 and very similar in "summer" and "winter", reflecting the location of wharves, terminals and anchorages, with visibly increased emissions from Halifax Seaport in "summer", where cruise ships are at berth.   Table 5

shows the calculated % increase in trace gas and PM emissions (according to the emission model in Tables 2 and 3) between "winter" and "summer", with $NO_x$ increasing by 4.1% and $SO_2$ by 3.2%, as well as the breakdown of this information by ship type, also indicating the percentage from passenger (i.e., cruise) ships in each season, which is 18% on average for all species in "summer" and 0.5% in "winter".   Cruise ship emissions are caused by their high auxiliary loads

while at berth (Table 2).   Somewhat surprisingly, the greatest proportion of $SO_2$ emissions in "winter" (Table 5) comes from tugs (26%), followed by container ships (25%), tankers (19%) and supply vessels (14%), while in "summer" this changes to tugs (24%), container ships (21%), passenger vessels (17%) and tankers (15%).   The proportions are very similar for $NO_x$ as well as for $CO_2$, CO, HC and PM in "summer" and "winter" (Figure 15, Table 5).   In total, the combined

contribution of tugs and coastal supply ships to total $SO_2$ and $NO_x$ emissions in Halifax Harbour was ~40% in "winter" and ~30% in "summer".   In our calculations all coastal (non-ocean going) ships have zero emissions at berth (speed = 0, auxiliary load = 0, Table 2), therefore, their high emissions are related to frequent arrivals, departures and maneuvering status.

**3.7 Comparison of shipping emissions to other sources**



Finally, we compare our calculated emissions to the largest stationary source emitter in the Halifax / Dartmouth area, which is the 500 MW Tufts Cove generating station, opposite the Richmond Terminal, in Dartmouth (Figure 13). In 2015 it accounted for 94% of stationary $SO_2$ emissions to Halifax / Dartmouth air according to the National Pollution Release Inventory (NPRI,

http://ec.gc.ca/inrp-npri/donnees-data/index.cfm?lang=En) and resulted in 1443 tonnes of emitted $SO_2$ (from January 2015 – December 2015), as compared to a maximum (based on the maximum permitted FSC of 0.1%) of 231 tonnes of $SO_2$ from shipping emissions (in a one year period from May 2015 – April 2016), not including municipal ferries and pleasure craft. While the shipping emissions are 6.2 times smaller, they are at the surface as opposed to from the three 152-m

chimneys at Tufts Cove, which makes their impact to local air quality greater, especially during the winter under reduced mixing layer heights. The power plant reports a wide spread of annual $SO_2$ emissions, i.e., only 25 tonnes in 2012 with an average of 1741 tonnes per year between 2006 and 2015 (inclusive). This is likely related to it being a dual-fuel plant, which burns either natural gas or heavy fuel oil, depending on price and availability. Vehicle emissions of $SO_2$ are expected

to be small because of stricter regulations on fuel sulfur content (0.008% m/m). Indeed, the Air Pollutant Emissions Inventory (APEI, http://www.ec.gc.ca/inrp-npri/donnees-data/ap/index.cfm?lang=En), which strives to account for all emissions, not just stationary source emissions, estimates 51 tonnes of $SO_2$ from all on-road road emission sources in the entire province in 2015.

In terms of stationary source $NO_x$ emissions Tufts Cove is also the dominant emitter in 2015 (91%) at 1784 tonnes (only 1277 tonnes in 2006 with an average of 2603 tonnes between 2006 and 2015, inclusive). Our calculated shipping emissions for the May 2015 – April 2016 one year period are 7544 tonnes, or 4.2 times higher than Tufts Cove emissions. The APEI estimates provincial 2015 emissions of $NO_x$ (as $NO_2$-equivalent mass) as 15636 tonnes from power

generation, 36975 from marine transportation and 13109 from all other transportation combined (including air and rail), together accounting for 93% of province-wide emitted $NO_x$. The population of Halifax Regional Municipality (HRM) represents 45% of the population of Nova Scotia. If we make the highly simplifying assumption that as much as 50% of provincial $NO_x$ emissions from all non-marine transportation (i.e., 6555 tonnes) can be attributed to the HRM then

our calculated shipping emissions are the greatest contributor to $NO_x$ emissions in the HRM, and in any case, of comparable magnitude to vehicle transportation.



## 4 Summary and Conclusions

A mobile open-path Fourier transform infrared spectrometer was set up in Halifax Harbour (Nova Scotia, Canada), an intermediate port integrated into the downtown core, to measure trace gas concentrations in the vicinity of marine vessels, in some cases with direct or near-direct marine combustion plume intercepts. The fuel sulfur content has been enforced at a maximum of 0.1% since August 2012 and the harbour is also a $NO_x$ emission control area. As Halifax is a small urban area (annual mean $NO_x$ levels of 18 ppbv in 2015), the relative positive perturbation to $O_3$ concentrations from emissions is smaller than over the background marine boundary layer, but not negligible. It already requires actions for preventing air quality deterioration under the Nova Scotia air zone management framework, driven by the Canadian Ambient Air Quality Standards (NSE Air Quality Unit, 2015). And while shipping-related annual mortality estimates are low in Nova Scotia as a result of a low population density, the concentration of shipping-related PM2.5 has been estimated in previous studies as comparable to other major global shipping routes. Moreover, a broader strategy of continued investment in port facilities to support provincial growth targets for trade activity, tourism and aquaculture exports further motivates the continued study of shipping emissions in our region, and in similar settings elsewhere, to establish concentration baselines as regulations on $NO_x$, $SO_x$ and PM emissions evolve in a protracted international legal process.

Our study is the first application of the OP-FTIR measurement technique to real-time, spectroscopic measurements of $CO_2$, $CO$, $O_3$, $NO_2$, $NH_3$, $CH_3OH$, $HCHO$, $CH_4$ and $N_2O$ in the vicinity of harbour emissions originating from a variety of marine vessels, and the first measurement of shipping emissions in the ambient environment along the eastern seaboard of North America outside of the Gulf Coast. The spectrometer, its active mid-IR source and detector were located on shore while the passive retroreflector was on a nearby island, yielding a 455-m open path over the ocean (910 m two-way). Atmospheric absorption spectra were recorded during day, night, sunny, cloudy and substantially foggy or precipitating conditions, with a temporal resolution of 1 minute or better. The retrievals are robust against a range of wet or precipitating weather conditions. A weather station was co-located with the retroreflector to aid in processing of absorption spectra and interpretation of results, while a webcam recorded images of the harbour once per minute. Trace gas concentrations were retrieved from spectra by the



MALT non-linear least squares iterative fitting routine. During field measurements (7 days in Jul-Aug, 2016; 12 days in Jan, 2017) Automatic Identification System information on nearby ship activity was collected manually from a commercial website and used to calculate emission rates of shipping combustion products ($CO_2$, CO, $NO_x$, HC, $SO_2$), which were then linked to measured

concentration variations using ship position and wind information.

Concentrations of $CO_2$ and all other retrieved gases except $O_3$ show various degrees of enhancement on July 13, August 16, January 30 and Feb 1 during broad periods (~9 hrs) of low wind speeds and suppressed mixing. $O_3$ is completely or near-completely titrated during these extended time periods. Our results compare well with a NAPS monitoring station ~1 km away,

pointing to the extended spatial scale of this effect, commonly found in much larger European shipping channels. CO shows the same broad enhancements, but also many relatively narrow enhancements on summer afternoons, related to pleasure craft in the harbour, unlike in some other studies that do not report CO signatures from individual ships. $NH_3$ concentrations show some enhancement when $CO_2$ is enhanced but are highly variable in both summer and winter, as is

HCHO; however, HCHO experiences stronger relative enhancements than $NH_3$, pointing to the proximity of more concentrated sources to the measurement location, which are possibly shipping-related. $CH_3OH$ shows strong enhancements correlated with times of suppressed mixing, like HCHO, but only in winter, with summer background concentrations slightly higher than winter. Amongst $NH_3$, HCHO and $CH_3OH$, it is methanol that shows the most correlation to rush hour

vehicle activities in winter measurements. Finally, $CH_4$ and $N_2O$ are elevated when $CO_2$ and CO are elevated, implying similar sources.

We assessed the spectral signatures of several other gases and find that the strong spectral interference from absorption by atmospheric water vapour makes accurate NO, $SO_2$, $HNO_3$ and HONO retrievals impossible under the path and concentration conditions in our study. HONO

was below detection limits at all times, even during the extended emissions accumulation periods on January 30, when water vapour interference was at a minimum. We also retrieved $C_2H_6$, $C_2H_4$, and $C_2H_2$ with mixed success. The three hydrocarbon species showed some accumulation during periods of low wind speed, however, retrievals require further work and independent measurement verification, beyond the scope of this study.

We calculated total marine sector emissions in Halifax Harbour based on a complete AIS dataset of ship activity during the cruise ship season (May – Oct 2015) and the remainder of the





year (Nov 2015 – Apr 2016) and found trace gas emissions (tonnes) to be on average 2.8% higher during the cruise ship season, when passenger ship emissions were found to contribute 18% of emitted $CO_2$, CO, $NO_x$, $SO_2$ and HC, and 0.5% off season for the same species.   Similarly calculated particulate emissions are 4.1% higher during the cruise ship season, when passenger

ship emissions contribute 18% of emitted PM (0.5% off season).   Tugs were found to make the biggest contribution to harbour emissions of trace gases in both cruise ship season (23% $NO_x$, 24% $SO_2$) and off season (26% of both $SO_2$ and $NO_x$), followed by container ships (25% $NO_x$ and $SO_2$ in off season, 21% $NO_x$ and $SO_2$ in cruise ship season), but then either cruise ships in third place in season or tankers in third place off season, both responsible for 18% of trace gas emissions.

While the concentrations of regulated trace gases measured by OP-FTIR as well as the nearby in situ NAPS sensors were well below maximum hourly permissible levels (80 ppb for $O_3$; 13 ppm for CO (8-hourly); 210 ppb for $NO_2$; 340 ppb for $SO_2$) at all times during the 19 measurement days, we find that AIS-based calculated shipping emissions of $NO_x$ over the course of one year are 4.2 times greater than those of a nearby 500 MW stationary source emitter and greater than or

comparable to all vehicle $NO_x$ emissions in the city.   Our findings highlight the need to accurately represent emissions of the shipping and marine sectors at intermediate ports integrated into urban environments.   With ever increasing spatial resolution in chemical air quality forecasting models, it is becoming feasible to model wharf and shipping channel activities as additional pseudo-stationary and pseudo-line sources.

### Acknowledgements

The authors are grateful for property access and logistical support from Halifax Seaport Authority, the Department of Fisheries and Oceans, Parks Canada, Saint Mary's University and Captain Albert Conrod of A&M Sea Charters for getting us to George's Island in January.   Ralf Pickart

(Nova Scotia Webcams) provided 1-minute resolution photos of Halifax Harbour, including the measurement open path.   Matt Seaboyer at Nova Scotia Environment provided 1-min resolution NAPS data during the measurement period.   Casey Hilliard provided one year of AIS data from Dr. Taggart's (WHaLE project) at Dalhousie University.   Ship AIS data was also collected manually from www.marinetraffic.com during field measurement periods.   The authors received

funding from NSERC, CFI, NSRIT, MEOPAR-NCE, Saint Mary's University and the Province of Nova Scotia.



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





Table 1. Spectroscopic retrieval parameters and detection limits for gases measured with an optical path of 913.7 m.

| Target Gas | Spectral window (cm$^{-1}$) | Interfering gases | 3σ noise level detection limit for 913.7 m path |
|---|---|---|---|
| $CO_2$[a] | 2030 – 2133 | $H_2O$, CO | 4.0 ppm |
| CO | 2080 – 2133 | $H_2O$, $CO_2$ | 1.1 ppb |
| $NO_2$ | 2910 – 2924.5 | $H_2O$[b], $CH_4$ | 7.2 ppb |
| $O_3$ | 1031.5 – 1063 | $H_2O$, $CO_2$, $NH_3$, $CH_3OH$ | 4.5 ppb |
| NH3 | 1031.5 – 1063 | $H_2O$, $CO_2$, $O_3$, $CH_3OH$ | 0.8 ppb |
| HCHO | 2745 – 2800 | $H_2O$[b], $CH_4$, $N_2O$ | 1.5 ppb |
| $CH_3OH$ | 1031.5 – 1063 | $H_2O$, $CO_2$, $NH_3$, $O_3$ | 0.9 ppb |
| $CH_4$ | 2900 – 2963 | $H_2O$, NO2, HCHO | 1.1 ppb |
| $N_2O$ | 2132.5 – 2235 | $H_2O$, $CO_2$, CO | 0.6 ppb |

[a] $CO_2$ residual de-weighted from 2074-2080 in cost function to skip known line mixing fitting error
[b] HITRAN 2004 database used for water in these specific retrievals, otherwise HITRAN 2012



Table 2. Assumed ship types and engine loads for emissions calculations, after EPA (2000).

| Ship types | | Power (kW) | | Load fraction (%) and Auxiliary Loads (kW) of Cruise status | | | |
|---|---|---|---|---|---|---|---|
| | | | | Fast Cruise (≥ 12 knots) | Slow Cruise (5 – 12 knots) | Maneuvering (0 – 5 knots) | Dockside (0 knot) |
| Ocean going | Passenger Ship | -3646 + 5.09×(DWT) [a] | | 80% | 20% | 10% | 0% |
| | | Auxiliary Loads (kW) | | 5000 | 5000 | 5000 | 5000 |
| | Bulk Carriers / Oil / Chemical Tankers | 6780 + 0.076× (DWT) | | 80% | 40% | 20% | 0% |
| | General Cargo | 2277 + 0.215× (DWT) | | 80% | 35% | 20% | 0% |
| | Container / RORO / Reefer / Vehicle Carrier | 1929 + 0.537× (DWT) | | 80% | 30% | 15% | 0% |
| | Auxiliary Loads for non-passenger ships (kW) | | | 750 | 750 | 1250 | 1000 |
| Non-ocean going | | Power (kW) | DWT (tonnes) | | | | |
| | Supply / Tender / Icebreaker | 3772 | 1000 | 80% | 40% | 20% | 0% |
| | Military | 3772 | 5000 | | | | |
| | Fishing | 827 | 500 | | | | |
| | Tugs / Pilot | 3190 | 200 | | | | |
| | Ferries | 1805 | 1000 | | | | |
| | Yachts | 1393 | 200 | | | | |
| | Auxiliary Loads for non-oceangoing ships (kW) | | | 0 | 0 | 0 | 0 |

[a] DWT = Deadweight

Table 3. Emission factors for ship emissions calculations, after EPA (2000). FSC is the Fuel Sulfur Content (% m/m), set to 0.1% as per applicable SECA regulations in the calculation region and time.

| Emitted species | Estimation equation | a [g/kWh] | -x [1] | b [g/kWh] |
|---|---|---|---|---|
| $CO_2$ | Emission Rate (g/kWh) = a × (Load Fraction)$^{-x}$ + b | 44.1 | 1 | 648.6 |
| CO | | 0.8378 | 1 | n/s [a] |
| $NO_x$[c] | | 0.1865 | 1.5 | 15.5247 |
| HC | | 0.0667 | 1.5 | n/s [a] |
| PM | | 0.0059 | 1.5 | 0.2551 |
| $SO_2$ | Emission Rate (g/kWh) = a × ( Fuel Consumption (g/kWh)   × FSC) + b<br>= a × ((14.12/Load Fraction + 205.717) × FSC) + b | 1.998 | n/a[b] | n/s [a] |

[a] n/s = not statistically significant

10    [b] n/a = not available

[c] $NO_x$ emission rate gives the $NO_2$-equivalent mass of emitted $NO_x$



Table 4. Ships in Halifax Harbour during field measurements at Halifax Seaport and their calculated emissions.

| Measurement | Summary of Ships | | | | Calculated Emissions (Tonnes) | | |
|---|---|---|---|---|---|---|---|
| | Type | Number | DWT (T) | Total Number | Gas | From All Ships | Per day |
| July 12 – 15, 2016 (3d, 2.5h) | Container/Cargo | 15 | 743,181 | 46 | $SO_2$ | 2.52 | 0.81 |
| | Oil Tanker | 5 | 150,271 | | $NO_x$[a] | 81.78 | 26.34 |
| | Vehicle carrier | 3 | 65,136 | | CO | 18.01 | 5.80 |
| | Navy | 4 | N/A | | $CO_2$ | 3962.01 | 1276.35 |
| | Tugs/Supply/Others | 19 | N/A | | HC | 3.41 | 1.10 |
| | | | | | PM | 1.49 | 0.48 |
| August 15 – 17, 2016 (2d, 6.5h) | Container/Cargo | 12 | 685,307 | 52 | $SO_2$ | 2.05 | 0.90 |
| | Oil Tanker | 2 | 51,633 | | $NO_x$[a] | 66.33 | 29.21 |
| | Vehicle carrier | 2 | 46,770 | | CO | 14.51 | 6.39 |
| | Navy | 2 | N/A | | $CO_2$ | 3225.35 | 1420.34 |
| | Tugs/Supply/Others | 34 | N/A | | HC | 2.62 | 1.15 |
| | | | | | PM | 1.20 | 0.53 |
| January 23 – February 3, 2017 (11d, 4h) | Container/Cargo | 28 | 1,299,474 | 65 | $SO_2$ | 5.09 | 0.45 |
| | Oil Tanker | 4 | 89,569 | | $NO_x$[a] | 163.71 | 14.66 |
| | Vehicle carrier | 2 | 31,848 | | CO | 37.94 | 3.40 |
| | Navy | 7 | N/A | | $CO_2$ | 8011.35 | 717.43 |
| | Tugs/Supply/Others | 24 | N/A | | HC | 6.98 | 0.63 |
| | | | | | PM | 2.98 | 0.27 |

[a] $NO_x$ is expressed as $NO_2$-equivalent mass

Table 5. Calculated marine sector emissions (tonnes) in Halifax Harbour in "summer" cruise ship season (May 2015 - October 2015) and "winter" non-cruise ship season (November 2015 - April 2016), based on detailed AIS ship type and activity data and the emission model from Tables 2 and 3.  Also shown are the % increase in emitted mass from "winter" (W) to "summer" (S), and the % of emissions due to passenger (primarily cruise ship) vessels.

| | $SO_2$ | | $CO_2$ | | CO | | $NO_x$[a] | | HC | | PM | |
|---|---|---|---|---|---|---|---|---|---|---|---|---|
| | W | S | W | S | W | S | W | S | W | S | W | S |
| Oil T. | 21.2 | 17.2 | 33394.5 | 27100.9 | 140.4 | 112.1 | 692.6 | 563.4 | 24.8 | 19.8 | 12.4 | 10.1 |
| Bulk Car. | 4.0 | 2.2 | 6309.4 | 3379.4 | 28.1 | 15.9 | 130.5 | 69.6 | 5.2 | 3.1 | 2.4 | 1.3 |
| Gen. Car. | 3.0 | 3.9 | 4650.2 | 6117.8 | 16.7 | 19.3 | 98.1 | 130.9 | 2.8 | 3.1 | 1.7 | 2.3 |
| Passenger | 0.6 | 20.4 | 914.4 | 32095.3 | 4.2 | 136.7 | 19.3 | 681.5 | 0.9 | 30.2 | 0.4 | 12.5 |
| Container | 28.5 | 24.7 | 44859.5 | 38777.3 | 213.2 | 184.6 | 920.4 | 795.3 | 40.8 | 35.3 | 16.8 | 14.6 |
| Ro-Ro Car. | 0.9 | 2.3 | 1466.2 | 3557.3 | 6.8 | 9.2 | 30.2 | 77.8 | 1.3 | 1.5 | 0.6 | 1.3 |
| Veh. Carr. | 2.4 | 1.7 | 3757.3 | 2691.7 | 17.6 | 10.7 | 77.4 | 56.5 | 3.4 | 2.0 | 1.4 | 1.0 |
| Fishing | 0.0 | 0.2 | 32.5 | 241.5 | 0.1 | 1.1 | 0.7 | 4.9 | 0.0 | 0.2 | 0.0 | 0.1 |
| Tug | 29.2 | 27.6 | 45899.4 | 43441.4 | 212.2 | 199.9 | 936.7 | 887.1 | 37.2 | 35.0 | 17.0 | 16.1 |
| Supply | 15.4 | 6.7 | 24197.8 | 10608.3 | 114.1 | 49.3 | 492.4 | 216.3 | 20.2 | 8.7 | 8.9 | 3.9 |
| Yacht | 0.0 | 1.2 | 18.6 | 1902.0 | 0.1 | 9.1 | 0.4 | 38.7 | 0.0 | 1.6 | 0.0 | 0.7 |
| Military | 8.4 | 9.3 | 13167.9 | 14567.3 | 62.0 | 68.6 | 268.0 | 296.5 | 11.0 | 12.1 | 4.9 | 5.4 |
| TOTAL | 113.7 | 117.3 | 178667.8 | 184480.1 | 815.3 | 816.5 | 3666.6 | 3818.4 | 147.6 | 152.5 | 66.5 | 69.2 |
| % Increase | | 3.2 | | 3.3 | | 0.1 | | 4.1 | | 3.4 | | 4.1 |
| % Passeng. | 0.5 | 17.4 | 0.5 | 17.4 | 0.5 | 16.7 | 0.5 | 17.8 | 0.6 | 19.8 | 0.5 | 18.0 |

[a] $NO_x$ is expressed as $NO_2$-equivalent mass





Figure 1. Location of Halifax Harbour within the Halifax Regional Municipality and the Canadian Maritime Provinces: Nova Scotia (NS), Prince Edward Island (PEI) and New Brunswick (NB). Also shown is the location of provincial/federal NAPS trace gas and aerosol monitoring, the measurement location (Halifax Seaport) and Saint Mary's University (SMU).

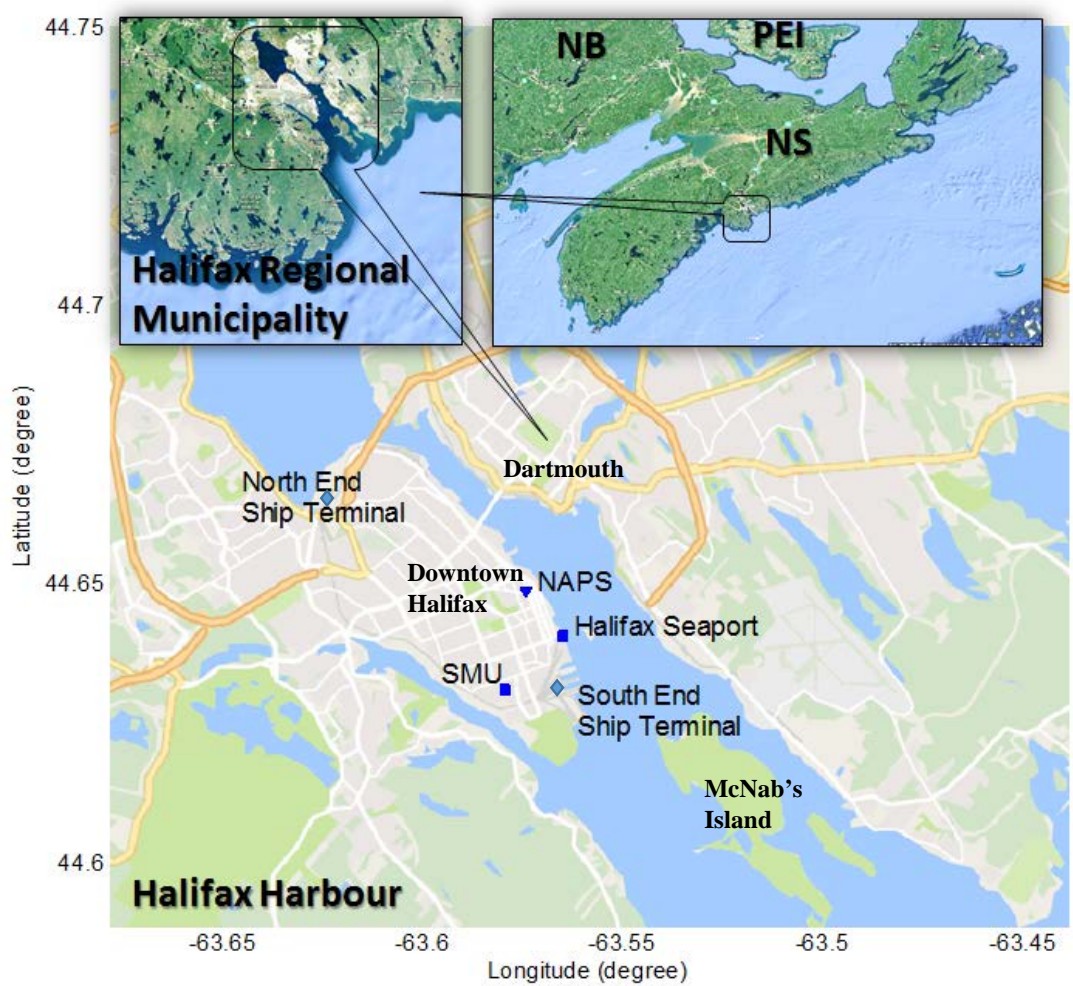





Figure 2. OP-FTIR system setup. (a) Spectrometer location with IR source, FTIR spectrometer, a single transmitting & receiving telescope, and detector.  Retroreflector location on lighthouse catwalk and weather station (not shown).  (b) Plan view of measurement geometry with sample speedboat crossing below open path.  Top inset:  Container Cargo vessel moving behind the island.  Bottom inset:  location of retroreflector (uninstalled) marked with red square.

5      (a)

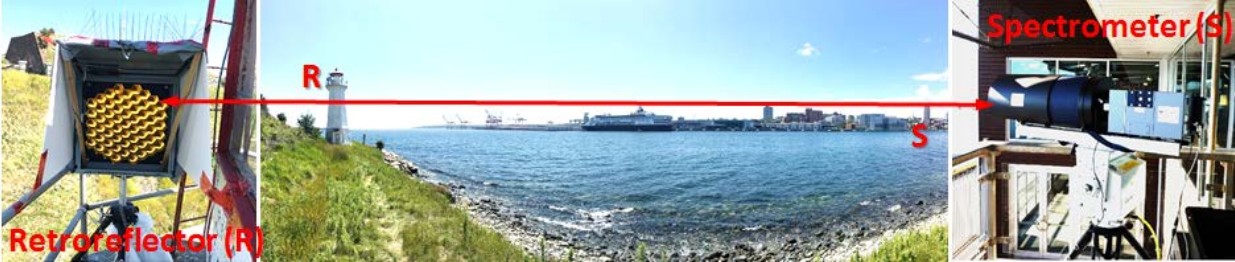

(b)

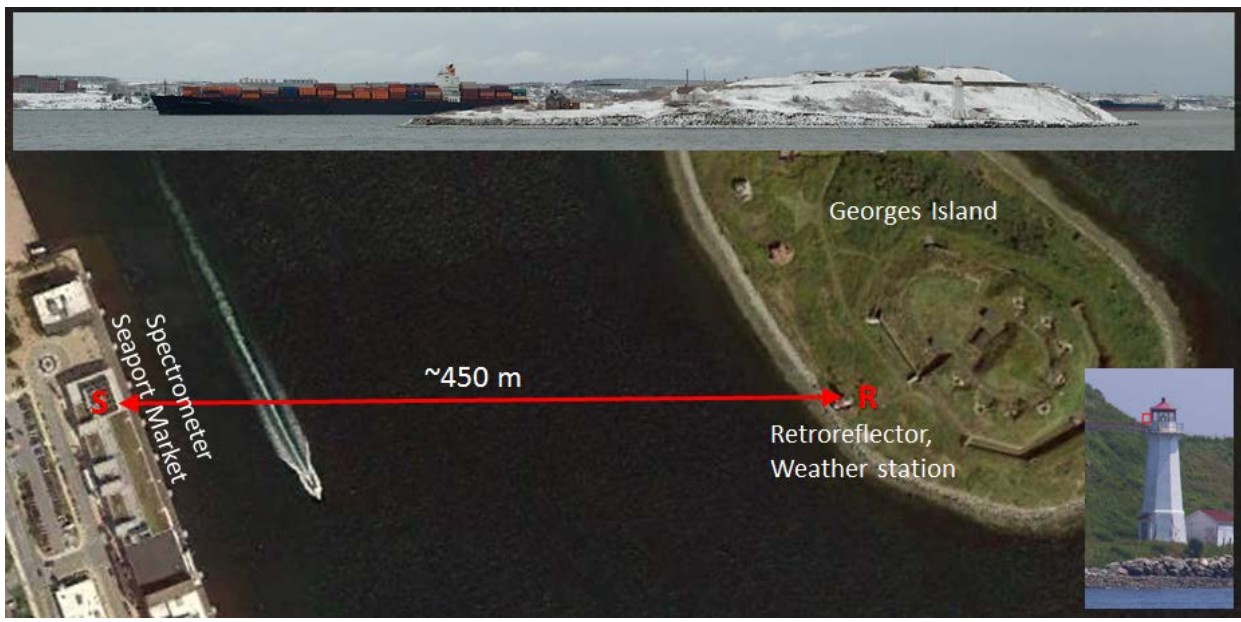





Figure 3. Effect of fog and rain on infrared signal intensity at 2100 cm-1 (bottom panel, arbitrary units) and retrieved O3 concentration (top panel) selected with and without a minimum signal threshold of 0.05 arbitrary units (a.u.).

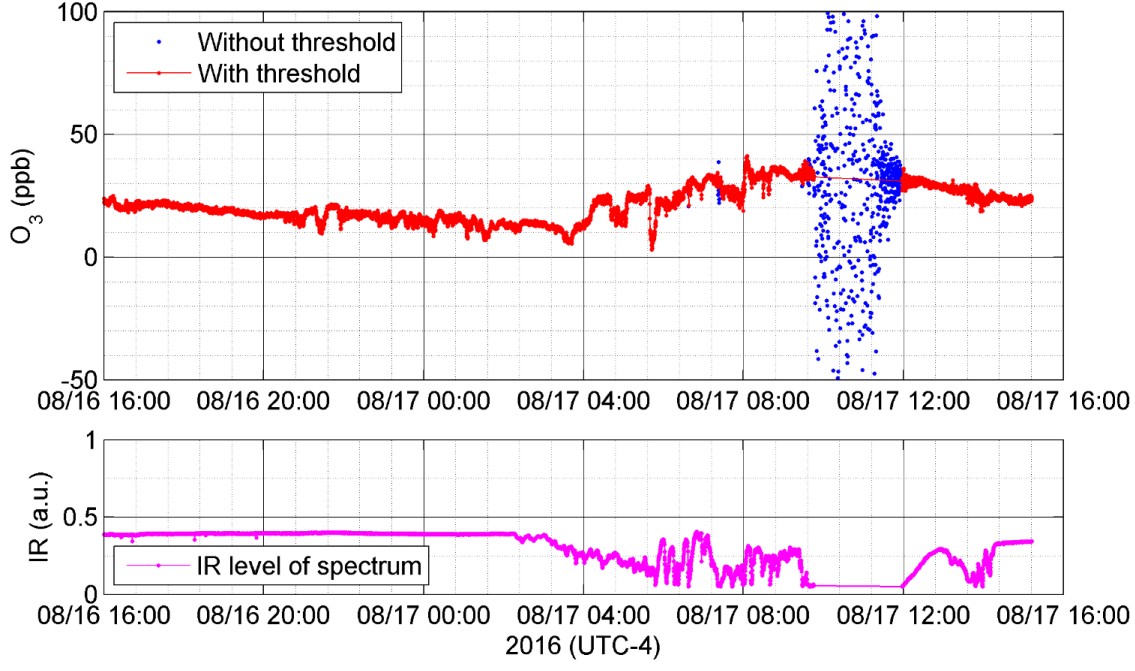

Figure 4. Distribution of ship activities during August 15-18 field measurements, not including ships not equipped with AIS transponders (small pleasure craft, municipal ferries).

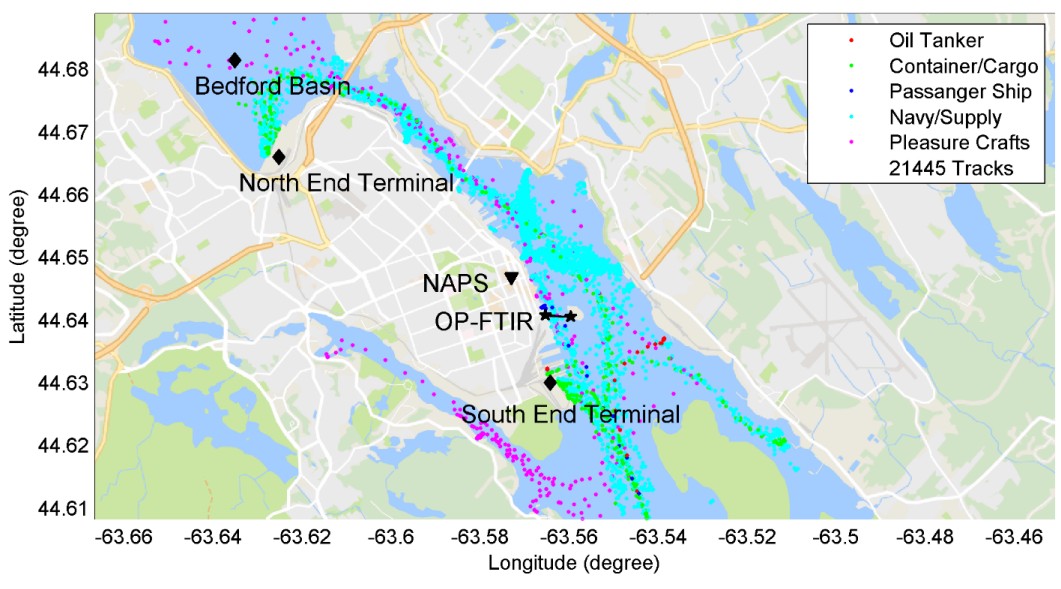



Figure 5. Retrieved concentrations of trace gases during summer (red) and winter (blue) field measurements.





Figure 6. Measured trace gas concentrations and main ship activities during extended emissions accumulation of July 13, 2016. Ship arrival (pink triangles), departure (blue triangles), and maneuvering (green dots) shown for main ships only.

(a) Track of Bulk Carrier with gas concentrations measured by OP-FTIR and wind information taken from SMU (1.8 km SW from open path, wind unavailable from island lighthouse on this day). HCHO panel also shows ship presence.

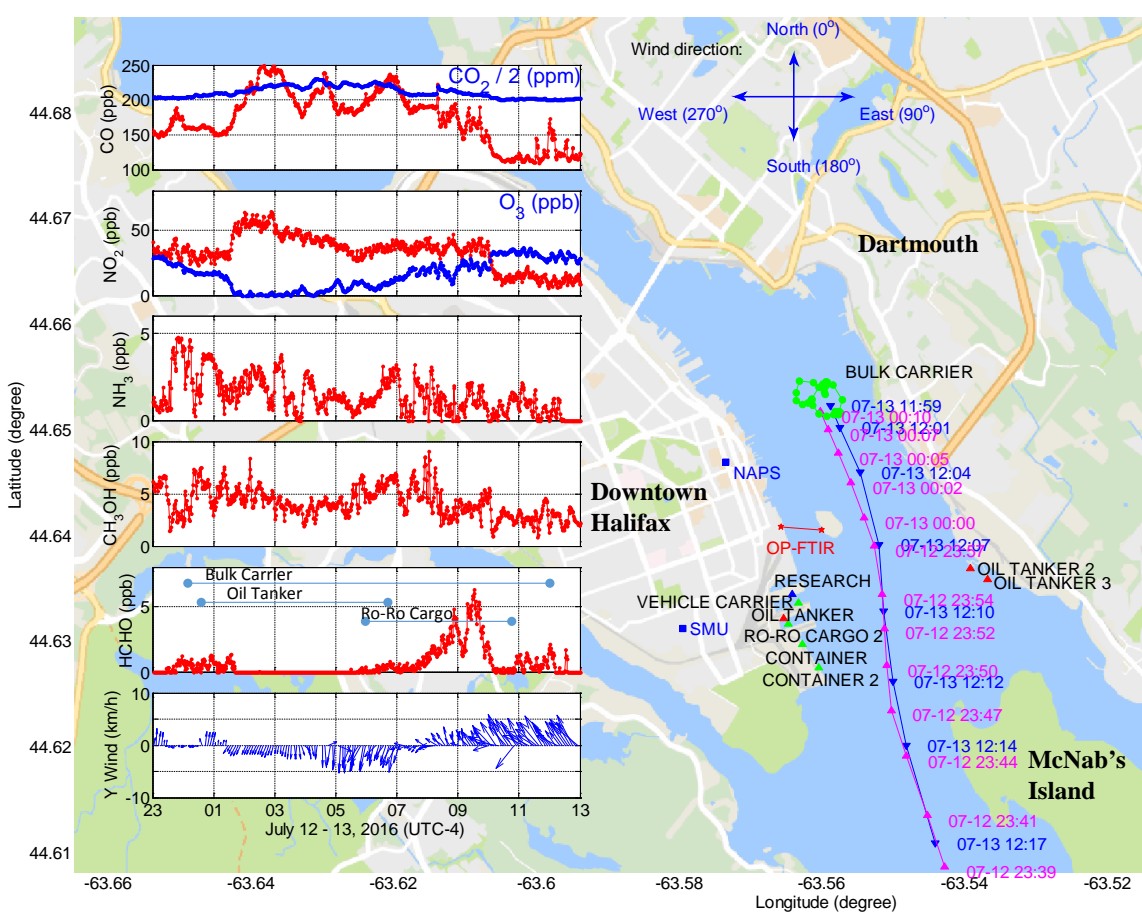

(b) Track of Oil Tanker

(c) Track of Ro-Ro Cargo

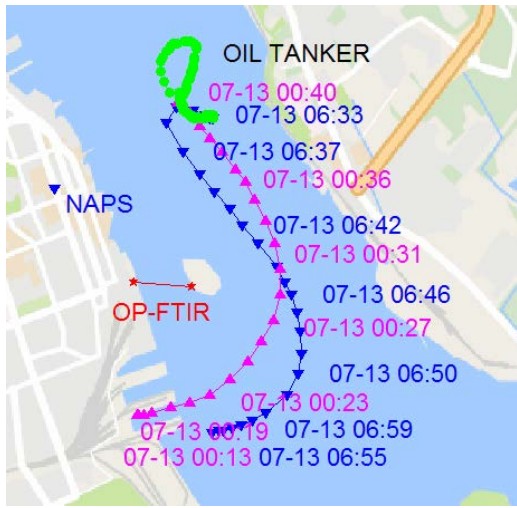

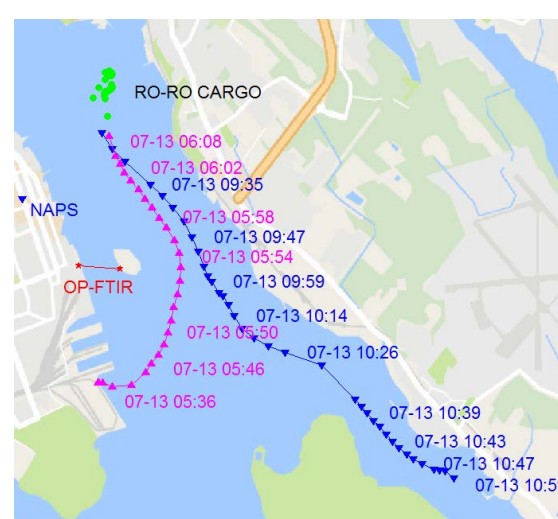



Figure 7. Calculated range of emissions for different operating modes of (a) Bulk Carrier (b) Oil Tanker and (c) Ro-Ro Cargo using the emissions model from Table 1 and 2. $NO_x$ represents $NO_2$-equivalent mass.

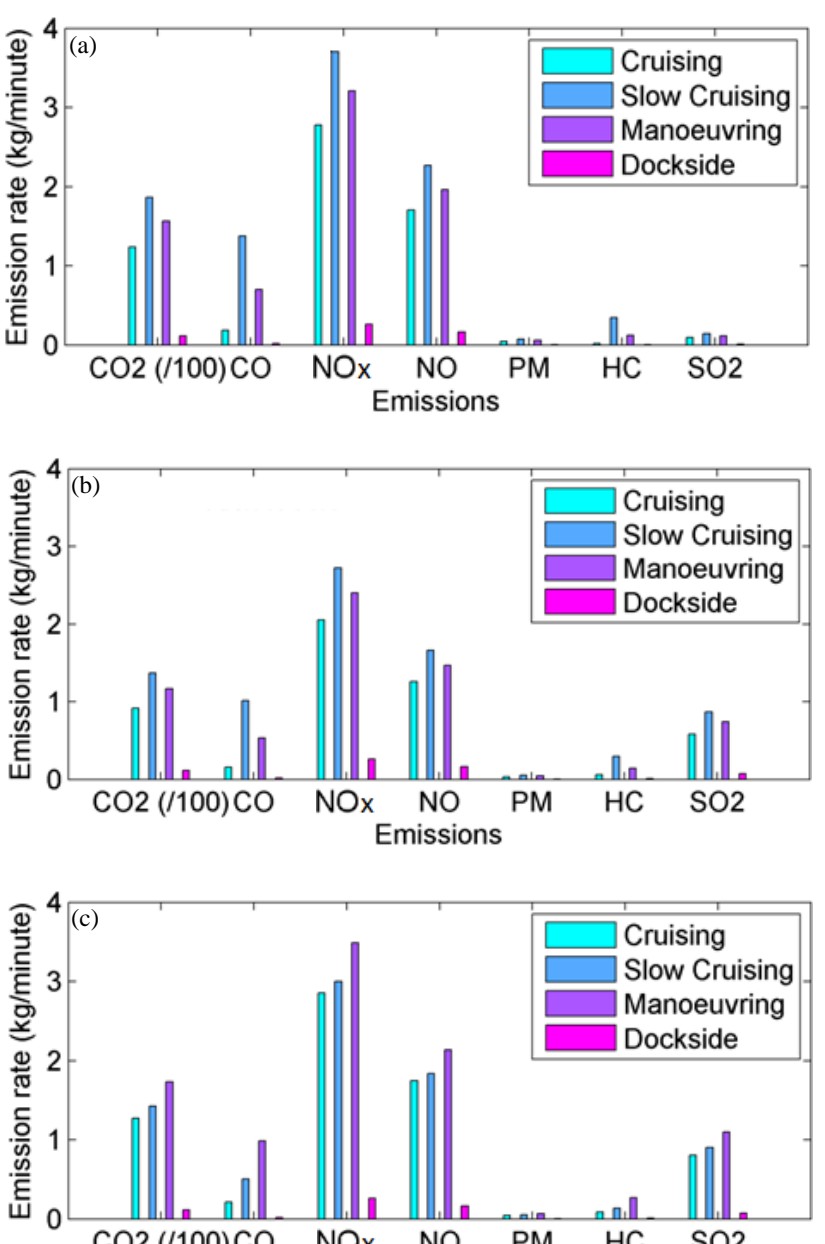



Figure 8. Detailed analysis of trace gas concentrations (a) under slowly changing winds (b) from 8:00 (time index 0) to 9:00 (time index 10) as emissions from two clusters of ships (c) were calculated based on their instantaneous AIS activity report at 8:20 (East ships: Warship = slow cruising; Oil Tanker 2 & 3 = dockside) and 8:50 (South Ships: dockside).



Figure 9. Measured trace gas concentrations during extended emissions accumulation on January 30, 2017. Wind direction (bottom most panel) before 12:00 is missing or highly variable because the wind speed is ~0 km/h.





Figure 10. Major ship activities during extended emissions accumulation on January 30, 2017, with only selected time stamps shown for increased clarity.



Figure 11. Comparison between trace gas concentrations measured in Halifax from January 23 – February 3, 2017 at NAPS station (in situ measurement, ~300 m from harbour, strongly influenced by vehicle traffic) and with OP-FTIR system ~1 km away (455 m open path measurement, spanning across harbour water, less influenced).  Zero values in NAPS measurements represent periods of calibration or no data (e.g., on Jan 30 for CO).  Bottom panel shows wind speed and direction during the time period, measured at retroreflector with OP-FTIR system.

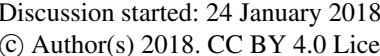





Figure 12. Combined effects of small and large ships in field measurements of CO, NO2 and O3 on July 13, 2016 under stable (SE) sea breeze (20 km/hr) winds.

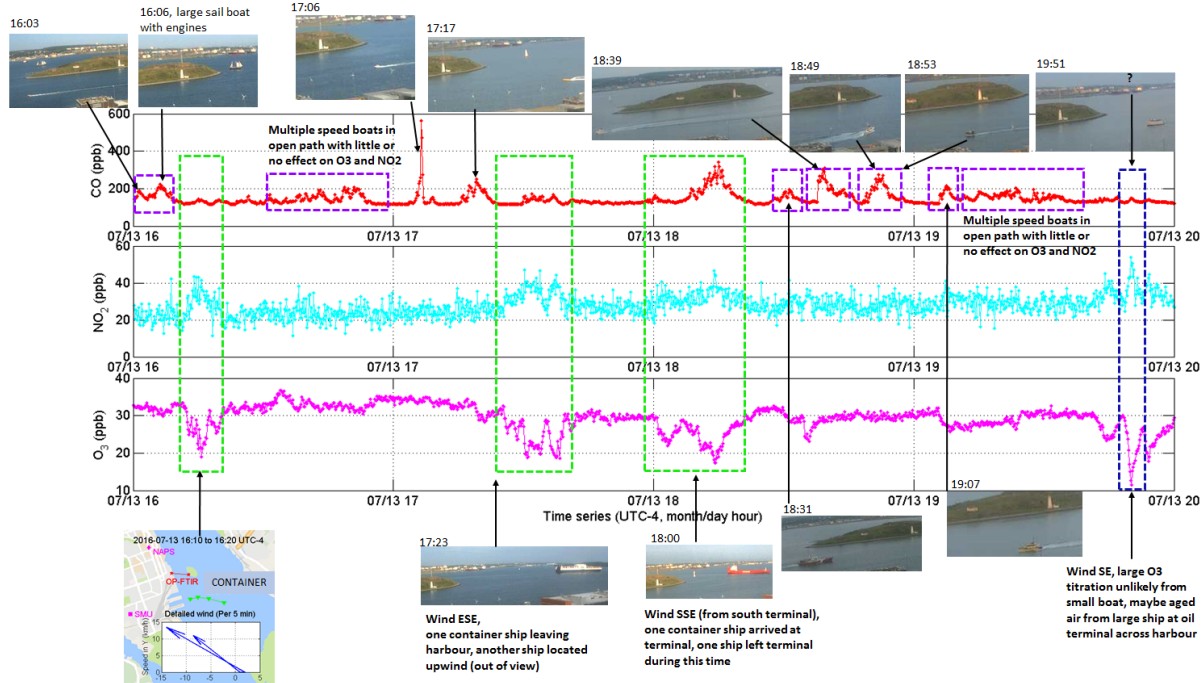





Figure 13. Calculated ship emissions (tonnes of NO$_x$ expressed as NO$_2$-equivalent mass, shown using unfilled but coloured contours) in Halifax Harbour based on AIS ship type and activity information during variable length summer (July only) and winter (January) measurement periods.

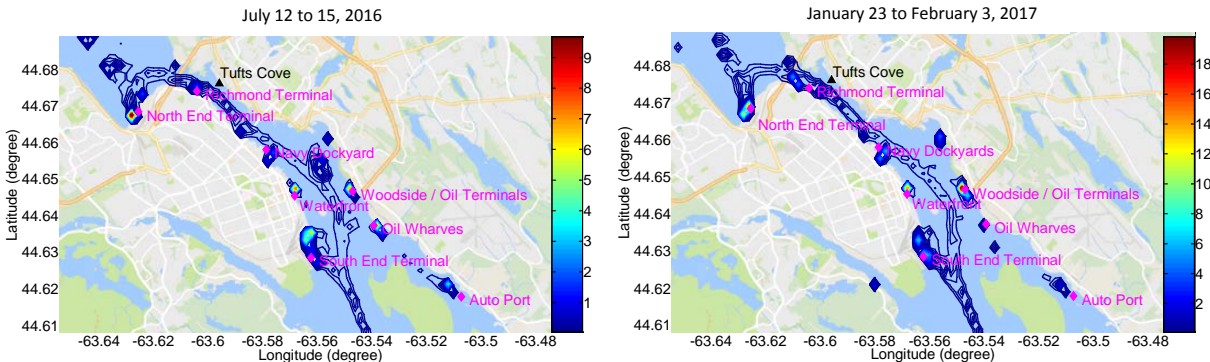

Figure 14. Proportion of NO$_x$ emissions from different ship types in "summer" (May 2015 to October 2015, left) and "winter" (November 2015 to April 2016, right).

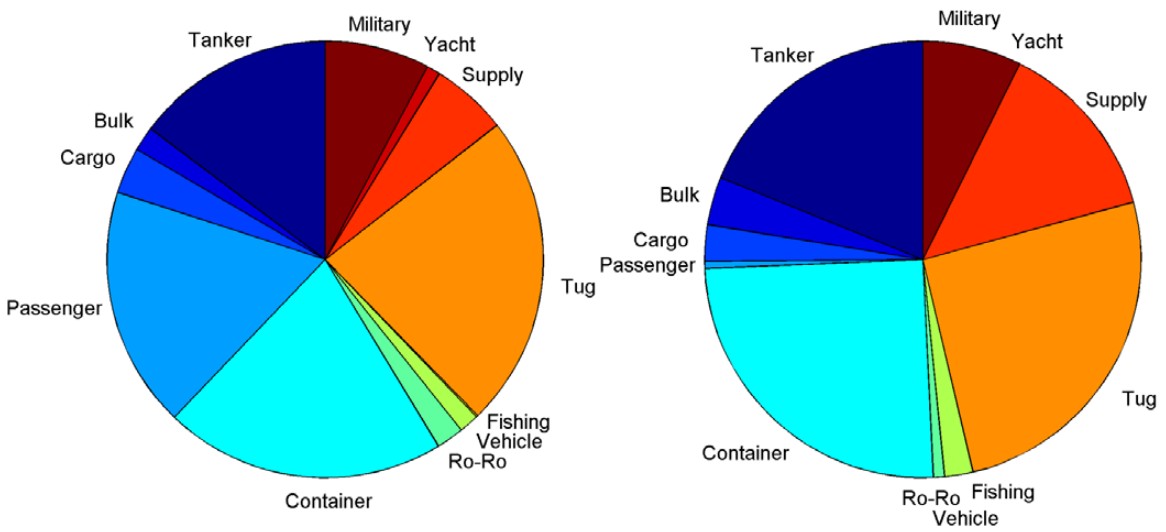

