# Peer review of "Characterization of trace gas emissions at an intermediate port"

_Atmospheric Chemistry and Physics, 2017_

## Referee Comment (RC1) · Anonymous Referee #1 · 14 Mar 2018

General Comments:

This is a very nicely written paper, detailing some interesting and novel work, and I thank the authors for their work and insight. The unique application of the OP-FTIR using the specific geography of Halifax Harbour and the direct identification of several features of ship plumes, as well as a discussion of the spatial impact of emissions from a medium-sized port in the Atlantic Canadian coastal environment, make this an important contribution to the scientific literature. The paper discusses a) the high-frequency detail of the field observations, b) the relevance of the observed trace gas perturbations due to ship emissions in relation to other major trace gas emitters in this coastal urban environment, and c) the broader global context of this issue, making it quite a thorough read. The paper identifies the impacts of both large ships (including pas-

senger, Container, Bulk Carrier, Cargo, Ro-Ro, Tankers, Navy) as well as – uniquely – small pleasure craft, on trace gas concentrations, and highlights the importance of further investigation into the impact of marine emissions in this region. The analysis and discussion is thorough, although as detailed below I would like to see quantitative correlations to support the discussion points. Qualitative time series correlation is compelling when it appears to makes sense, but for robust take-away messages more quantitative analysis should be considered. I have made a few minor suggestions throughout to improve the clarity, and I support the publication of this work in ACP.

Specific Comments:

Abstract

Why do passenger ships contribute 0.5% to emissions in off-season? (when there are no cruise ships between November and April)

Introduction

Scope of Introduction is appropriate and well-researched.

p3 lines 8,9,10 seem to contain a contradiction with respect to whether SOx is projected to increase or decrease: please clarify.

P7 lines 22-27: Are these two sentences relevant? The mention of LRT seems to pull focus from the point of the paper, but are you setting up for the suggestion that the Gibson LRT number includes what should rightfully be included in the shipping component? If so, I did not find a continuation of the discussion later in the paper. Either way, these two sentences detract from the well-made points of this paper and I would remove them.

Fig. 1: Whereas your paper is situated in an international context, I suggest making your "widest shot" an image of Canada, rather than the Maritime Provinces.

Results and Discussion

p14 line 13 – it is too bad the SO2 signature showed interference and could not be reported, this would be an interesting measurement given the recent SECA regulations. Do you have any reliable data?

I appreciate the discussion of what was not reported.

P14 line 31 – This sentence is confusing . . . does it suggest that plant respiration of CO2 would cause an enhancement of CO2? Regardless, I'm not sure a comment regarding plant respiration is relevant here. Would a citation for your previous work (p15 line2) help to clarify?

P15, line 22 – there are what appear to be anti-correlated variations in CO and NO2 at about 03:40 through to 07:00, when winds are directly from the north and NNE (Figure 6a). To what is this attributed, if not to the Bulk Carrier and Oil Tanker north of the OP-FTIR path? Quantitative correlations would be very helpful in understanding the relationships between constituents, and the times at which they cease to be correlated or anti-correlated, especially when attributing their causes (lines 22-25). Did you perform any correlations between the data series (during specific times of interest)?

P15, line 31 – This is a broad statement that may be considered speculative. I suggest ". . .wind was north / north-east MAY HAVE BEEN (or WAS LIKELY) CAUSED by 1) the Ro-Ro . . .."(author should replace "was caused" by one of the capitalized phrases).

P17 line 8 – did you calculate correlations between constituents for specific time frames, or is this statement based on eyeballing the time series'? Quantitative relationships between constituents for case studies where ship plumes were clearly observed would be a useful addition to your analysis.

P17 line 24 – quantitative correlations to support this statement could be added.

Figs. 9&11 – suggest to represent the wind speed and direction with a vector as you have done in Fig. 6.

Fig 11 – the agreement between the FTIR and the NAPS measurements is compelling,

and the discrepancies interesting. I agree that it warrants further investigation, especially considering the NAPS site is long-running and more analysis could be done on historical data where meteorological conditions indicate that the influence of ship emissions could be fairly wide-spread, as they are on January 30. This could further highlight the influence of ship plumes on the population of the port city, without embarking on a new field study.

P21 lines 29-31 – I would say that I do not agree that, by sight, the spikes occur mostly in the late afternoon and early evenings. It does stand to reason that pleasure craft may be more active later in the day, at least on weekdays. However, on two (Jul 14, Aug 16) of the three days (Jul 13, 14, Aug 16) where you have full days of data, the CO spikes appear to begin around 6am and persist until midnight. Do the earlier spikes follow the same pattern of enhancements/depletions?

P23 line 19, what accounts for the 0.5% in winter, as there are no cruise ships between Nov and April?

Technical Comments

P7 line 18 - "...to contribute between <10% (Hingston 2005) and ~30% (Phinney et al., 2006) OF AMBIENT CONCENTRATIONS in Halifax ..." (author should add the capitalized phrase if that is the correct interpretation; otherwise clarify)

P9 line 5 – add space between "corner" and "array"?

P9 line 5 – the word "if" should be "is".

P. 20 line 24 – take out "In summary, "

Fig 8 – add date to caption (suggest after "(time index 10)" )

Fig. 10 – it is difficult to understand which paths are attributable to which ships. Suggest to add a black connector from the description/time to the coloured path, or add a legend.

Fig 12 / p22 line 14 – The events at 20:00 are mentioned; do you mean 19:51? Whereas the analysis in this section refers to events happening on the order of minutes, the text should be corrected to read "19:51".

[Figure]

---

## Referee Comment (RC2) · Anonymous Referee #2 · 16 Mar 2018

This manuscript introduces a new method for monitoring trace gas emissions in a port area using open path Fourier transport infrared spectrometry over a distance of 455 m. This represents an important development in monitoring of trace gas emissions, although applications in other areas may be limited by the availability of source and detector locations at a suitable distance apart. I recommend that the authors consider the following comments before publication: 1. One significant omission in the measurement capability is SO2: estimates of SO2 emissions are based solely on AIS information. Since SO2 has been treated by the International Maritime Organisation as the highest priority pollutant gas, this limitation should be stated clearly in the Abstract and the Conclusions. 2. Page 11, lines 6-8: the "arbitrary units" appear to be treated as an absolute measure in assessing the validity of the ozone concentration measure-

ments. I assume that the units used for IR intensities are not in fact arbitrary, but rather calculated using a fixed procedure. It should be made clear that these numbers are not in fact arbitrary. 3. Page 12, line 22: Am I correct in understanding that the emission rates were calculated for all the gases in Table 3? Please make clear which rates were calculated. 4. The manuscript includes a detailed analysis of emissions in relation to ship movements on 2 separate days (sections 3.2 and 3.3). I consider that one of these analyses is sufficient to demonstrate the capabilities of the method. 5. Page 24, lines 20-26: it is stated that Tufts Cove is the dominant stationary NOX source at around 2000 tonnes per year, yet the provincial total for power generation is 15 636 tonnes per year. Does this mean that there are much larger sources elsewhere in Nova Scotia? The provincial totals seem very high compared to the Halifax emissions given that Halifax accounts for almost half of the population. Please provide some more background to these figures. 6. The manuscript necessarily includes many acronyms. It would help the reader to collect the acronym definitions in a Table. Some minor editorial points: 1. Page 8, line 5 and Table 1: some figures not subscripted 2. Page 9, line 5: "is" not "if" 3. Page 13, lines 27 and 28: the normal convention is to use T for temperatures in kelvin and t for temperatures in degrees Celsius. 4. Figure 6a: do the concentration plots follow the colour scheme red for arrival, blue for departure? Please make clear in the figure legend.

---

## Author Comment (AC1) · 9 Jul 2018

The attached document contains responses to Referee #1 and #2, a revised and tracked version of the manuscript as well as the tables and figures file.

Please also note the supplement to this comment:
https://www.atmos-chem-phys-discuss.net/acp-2017-1153/acp-2017-1153-AC1-supplement.pdf

---

## Author Response (AR1)

**General Comments:**

*This is a very nicely written paper, detailing some interesting and novel work, and I thank the authors for their work and insight. The unique application of the OP-FTIR using the specific geography of Halifax Harbour and the direct identification of several features of ship plumes, as well as a discussion of the spatial impact of emissions from a medium-sized port in the Atlantic Canadian coastal environment, make this an important contribution to the scientific literature. The paper discusses a) the high-frequency detail of the field observations, b) the relevance of the observed trace gas perturbations due to ship emissions in relation to other major trace gas emitters in this coastal urban environment, and c) the broader global context of this issue, making it quite a thorough read. The paper identifies the impacts of both large ships (including passenger, Container, Bulk Carrier, Cargo, Ro-Ro, Tankers, Navy) as well as – uniquely – small pleasure craft, on trace gas concentrations, and highlights the importance of further investigation into the impact of marine emissions in this region. The analysis and discussion is thorough, although as detailed below I would like to see quantitative correlations to support the discussion points. Qualitative time series correlation is compelling when it appears to makes sense, but for robust take-away messages more quantitative analysis should be considered. I have made a few minor suggestions throughout to improve the clarity, and I support the publication of this work in ACP.*

We thank Referee #1 for their constructive general and specific comments. We have refined our use of the term 'correlation' in discussing co-varying trace gas concentrations, as outlined in more detail below.

**Specific Comments:**

**Abstract**

*Why do passenger ships contribute 0.5% to emissions in off-season? (when there are no cruise ships between November and April)*

There were in fact 2 cruise ships in port already in April in 2017 (1 in April in 2016), out of 180 total (136 total in 2016). We noted this in the manuscript Abstract and Conclusions for greater clarity.

**Introduction**

*Scope of Introduction is appropriate and well-researched.*

Thank you – we worked hard on it.

*p3 lines 8,9,10 seem to contain a contradiction with respect to whether SOx is projected to increase or decrease: please clarify.*

Reworded to distinguish growth of emissions from the entire shipping sector even as individual ship emissions decline due to fuel and engine regulations.

*P7 lines 22-27: Are these two sentences relevant? The mention of LRT seems to pull focus from the point of the paper, but are you setting up for the suggestion that the Gibson LRT number includes what should rightfully be included in the shipping component? If so, I did not find a continuation of the discussion later in the paper. Either way, these two sentences detract from the well-made points of this paper and I would remove them.*

Re-reading again we see that the LRT discussion indeed detracts from the local sources discussion.  We think it is important to discuss the very few studies of Halifax but have made changes to focus only on shipping-related aerosol numbers, which are given in both papers, alongside LRT contributions:  3.4% PM2.5 mass (Gibson) and 9.1% PM2.5 mass (Jeong).

*Fig. 1: Whereas your paper is situated in an international context, I suggest making your "widest shot" an image of Canada, rather than the Maritime Provinces.*

We have expanded the widest shot to include Eastern Canada and Northeast United States.

**Results and Discussion**

*p14 line 13 – it is too bad the SO2 signature showed interference and could not be reported, this would be an interesting measurement given the recent SECA regulations. Do you have any reliable data?*

Unfortunately, water interference is severe where the $SO_2$ spectral signature is the strongest, and $SO_2$ concentrations have dramatically decreased due to successful regulations.  We are continuing to systematically study the sensitivity of the OP-FTIR $SO_2$ retrieval to water interference and other retrieval parameters and intend to expand on this in a future publication outlining when and where the retrieval may be successful using the technique.

*[p14 line 13 cont'd] I appreciate the discussion of what was not reported.*

Thank you – we included information on species that could be possible to retrieve from measured spectra for other investigators, specifically when working in less humid environments or under enhanced primary target concentration conditions.

*P14 line 31 – This sentence is confusing . . . does it suggest that plant respiration of CO2 would cause an enhancement of CO2? Regardless, I'm not sure a comment regarding plant respiration is relevant here. Would a citation for your previous work (p15 line2) help to clarify?*

We clarified this sentence to include the reasoning for considering plant respiration, i.e., Halifax is surrounded by forests and the measurement period is in the peak of the growing season. We also noted that our forest data is currently unpublished and from only ~12 km away (at Lake Major).

*P15, line 22 – there are what appear to be anti-correlated variations in CO and NO2 at about 03:40 through to 07:00, when winds are directly from the north and NNE (Figure 6a). To what is this attributed, if not to the Bulk Carrier and Oil Tanker north of the OP-FTIR path?*

Yes, there is a period from 3:40 – 4:40 in which CO is increasing while $NO_2$ is slowly decreasing. We offered an explanation for this process on lines 22-25 in terms of a limit to the conversion of primary emitted NO to $NO_2$ via $O_3$ titration (once $O_3$ is used up), and suggested other dark processes of $NO_2$ removal by conversion to $NO_3$, $N_2O_5$, $HNO_3$ as well as the heterogeneous conversion of $NO_2$ to HONO and $HNO_3$. We also noted the complication of twilight beginning at 4:06 and sunrise at 4:41. From 5:30 to 7:00 CO and NO2 are rising together again.

*[P15, line 22 cont'd] Quantitative correlations would be very helpful in understanding the relationships between constituents, and the times at which they cease to be correlated or anti-correlated, especially when attributing their causes (lines 22-25). Did you perform any correlations between the data series (during specific times of interest)?*

Indeed, we performed extensive correlation calculations on various short and long (minutes to hours) time periods in an attempt to calculate ship emission factors, however, given the variable city background and path-averaging of signatures, we abandoned this line of work because the emission factors were very sensitive to the time periods chosen, and thus could not easily be attributed to single ship sources but rather only to city-wide emissions. This kind of emission factor is not directly comparable to other reported values for specific ship types.

While we agree that correlation factors would add a quantitative dimension to the analysis, they will not change the main conclusions of the paper, and we hesitate to add such numbers so as to not over-interpret the complex data (i.e., a 450-m path average of the contribution of multiple sources under light winds and over sunrise). We are working on increasing the sensitivity of our current setup and have field work scheduled in July/August 2018, where we hope to detect more species (e.g., NO) with less noise (e.g., $NH_3$, $CH_3OH$, HCHO) over a longer time period, where correlations may be more informative.

*P15, line 31 – This is a broad statement that may be considered speculative. I suggest ". . .wind was north / north-east MAY HAVE BEEN (or WAS LIKELY) CAUSED by 1) the Ro-Ro . . .."(author should replace "was caused" by one of the capitalized phrases).*

Agreed – changed to "was likely caused by".

*P17 line 8 – did you calculate correlations between constituents for specific time frames, or is this statement based on eyeballing the time series'? Quantitative relationships between*

*constituents for case studies where ship plumes were clearly observed would be a useful addition to your analysis.*

The positive correlation between HCHO, CO and $NO_2$ (and negative correlation with $O_3$) from 8:15 to 9:15 was inferred from eyeballing the time series. Rather than referring the reader to two separate figures to assess the correlation, we dropped the reference to Figure 8a (which stops at 9:00 anyway) and added a box around the event in question in Figure 6a, which makes the correlation alluded to in the text clear. We also changed to wording in the text from "positively correlated with" to "also associated with an enhancement of".

*P17 line 24 – quantitative correlations to support this statement could be added.*

We corrected a mistake on this line: we meant to highlight that there is a correlated rise in $NH_3$ (not also $CH_3OH$) and CO (not $CO_2$) at 1:20 am, the time that changing winds bring CO- and $NO_2$- rich air from the direction of the docked ships to the open path.

*Figs. 9&11 – suggest to represent the wind speed and direction with a vector as you have done in Fig. 6.*

We actually had it done both ways (!) but found the scalar representation more informative / more clear to read in the case of Figs 9 & 11.

*Fig 11 – the agreement between the FTIR and the NAPS measurements is compelling, and the discrepancies interesting. I agree that it warrants further investigation, especially considering the NAPS site is long-running and more analysis could be done on historical data where meteorological conditions indicate that the influence of ship emissions could be fairly wide-spread, as they are on January 30. This could further highlight the influence of ship plumes on the population of the port city, without embarking on a new field study.*

Agreed. We are working on the technical aspect of FTIR vs. in situ bias in a repeat field campaign in July/August 2018 (as noted above), also using co-located in situ measurements on either end of the open path to have a more direct comparison.

*P21 lines 29-31 – I would say that I do not agree that, by sight, the spikes occur mostly in the late afternoon and early evenings. It does stand to reason that pleasure craft may be more active later in the day, at least on weekdays. However, on two (Jul 14, Aug 16) of the three days (Jul 13, 14, Aug 16) where you have full days of data, the CO spikes appear to begin around 6am and persist until midnight. Do the earlier spikes follow the same pattern of enhancements/depletions?*

We agree that narrow spikes (defined in the text as ~1-15 minutes) can and do happen at times other than the afternoon, too. In the text we state that they are "especially" prevalent in late afternoon and early evenings. Indeed, all measurement days were weekdays (July 13/14 were Wed/Thu, while Aug 15/16 were Mon/Tue). With the scale of Figure 5b being quite zoomed out, it is not possible to see that the frequency of narrow spikes is much greater in the

afternoon and evening than in the morning on July 14. For the same reason of scale it is hard to see that the 6 AM spike on Aug 16 is in fact an extended event lasting over an hour, with narrow spikes being most prevalent in the afternoon and evening.

In looking at Figure 5 in detail again we uncovered an error where the y-tick labels were incorrect in Figure 5b (CO), 5d ($O_3$), 5h ($CH_4$) and 5i ($N_2O$). We fixed these and they do not affect the discussion because this was a late-stage figure editing error. The CO and $O_3$ y-axis labels have doubled, while $CH_4$ and $N_2O$ max y-value labels have been scaled (up) from 2.0 to 2.2 and from 340 to 360, respectively.

*P23 line 19, what accounts for the 0.5% in winter, as there are no cruise ships between Nov and April?*

There were in fact 2 cruise ships in port already in April in 2017 (1 in April 2016), out of 180 total (136 total in 2016). We noted this in the manuscript Abstract and Conclusions for greater clarity.

**Technical Comments**

*P7 line 18 - ". . .to contribute between <10% (Hingston 2005) and ~30% (Phinney et al., 2006) OF AMBIENT CONCENTRATIONS in Halifax . . ." (author should add the capitalized phrase if that is the correct interpretation; otherwise clarify)*

Thank you for pointing this ambiguity out. Hingston compiled a bottom up inventory of shipping emissions while Phinney estimated the shipping contribution to ambient concentrations based on a wind sector analysis. We clarified this in the text.

*P9 line 5 – add space between "corner" and "array"?* Done

*P9 line 5 – the word "if" should be "is".* Done

*P. 20 line 24 – take out "In summary, "* Done

*Fig 8 – add date to caption (suggest after "(time index 10)" )* Added "on July 13" before (a)

*Fig. 10 – it is difficult to understand which paths are attributable to which ships. Suggest to add a black connector from the description/time to the coloured path, or add a legend.*

Agreed. Expanded the caption to explain colors and time stamp positioning.

*Fig 12 / p22 line 14 – The events at 20:00 are mentioned; do you mean 19:51? Whereas the analysis in this section refers to events happening on the order of minutes, the text should be corrected to read "19:51".*

Yes – corrected to 19:51.
*This manuscript introduces a new method for monitoring trace gas emissions in a port area using open path Fourier transport infrared spectrometry over a distance of 455 m. This represents an important development in monitoring of trace gas emissions, although applications in other areas may be limited by the availability of source and detector locations at a suitable distance apart. I recommend that the authors consider the following comments before publication:*

*We thanks Referee #2 for their constructive comments.*

*1. One significant omission in the measurement capability is SO2: estimates of SO2 emissions are based solely on AIS information. Since SO2 has been treated by the International Maritime Organisation as the highest priority pollutant gas, this limitation should be stated clearly in the Abstract and the Conclusions.*

As in our reply to Referee #1, we note that water interference is severe where the $SO_2$ spectral signature is the strongest, and also that $SO_2$ concentrations have dramatically decreased due to successful regulations. This is a discussion already included in the manuscript (Section 3.1, where we discuss unsuccessful retrievals). Nevertheless, we are continuing to systematically study the sensitivity of the OP-FTIR $SO_2$ retrieval to water interference and other retrieval parameters and intend to expand on this in a future publication (e.g., in JQSRT) outlining when and where the retrieval may be successful using the technique. We have added this comment also in Section 3.1. As such, we think it is too early to make a general statement about the capability of the overall OP-FTIR technique with respect to $SO_2$ based solely on our particular measurement conditions in Halifax. In much drier/colder conditions with higher $SO_2$ concentrations the technique may well be applicable, and, furthermore, there are relatively straightforward technical modifications to explore which could make it more sensitive under all conditions.

*2. Page 11, lines 6-8: the "arbitrary units" appear to be treated as an absolute measure in assessing the validity of the ozone concentration measurements. I assume that the units used for IR intensities are not in fact arbitrary, but rather calculated using a fixed procedure. It should be made clear that these numbers are not in fact arbitrary.*

The uncalibrated units describing the intensity of the spectrum are commonly referred to as arbitrary units in FTIR spectroscopy because they are not calibrated radiances ($W/m^2/sr$) and cannot be compared directly even between two similar instruments measuring the same atmospheric path on account of the differing (but linear) response functions of the two instruments. Nevertheless, in our setup a threshold of 0.05 in spectral intensity is meaningful

as described, i.e., in that it leads to stable retrievals as judged by the RMS of the spectral fits. We added a comment in the manuscript noting that the threshold will be different in systems with a different response. Finally, the retrievals are performed on a calculated transmittance spectrum, where only relative absorption levels are important, as opposed to the (unknown) absolute radiance.

*3. Page 12, line 22: Am I correct in understanding that the emission rates were calculated for all the gases in Table 3? Please make clear which rates were calculated.*

Yes. Added reference to Table 3.

*4. The manuscript includes a detailed analysis of emissions in relation to ship movements on 2 separate days (sections 3.2 and 3.3). I consider that one of these analyses is sufficient to demonstrate the capabilities of the method.*

The discussion of the winter event is half the length of the discussion of the summer event. The winter measurement is more favourable spectroscopically due to lower water vapour content, and presents, e.g., CH$_3$OH related to winter activities. The summer measurement is less favourable, but captures the signatures of small pleasure craft, the effect of a higher sampling rate on repeatability and measurement scatter, and the broad differences due to different summer sources and summer insolation driving photochemistry. As such, both time periods are needed to characterize trace gas emissions at an intermediate port.

*5. Page 24, lines 20-26: it is stated that Tufts Cove is the dominant stationary NOX source at around 2000 tonnes per year, yet the provincial total for power generation is 15 636 tonnes per year. Does this mean that there are much larger sources elsewhere in Nova Scotia? The provincial totals seem very high compared to the Halifax emissions given that Halifax accounts for almost half of the population. Please provide some more background to these figures.*

Indeed, Tufts Cove (500 MW) represents only 20% of Nova Scotia's power generating capacity (2500 MW), 55% of which is currently achieved by burning coal elsewhere (Tufts Cove burns heavy fuel oil or natural gas, as noted). This increases the relative importance of shipping emissions in the city, which we noted at the end of Section 3.7.

*6. The manuscript necessarily includes many acronyms. It would help the reader to collect the acronym definitions in a Table.*

We eliminated acronyms used only once where this did not impact readability and we collected all other acronyms into Appendix A, noting where acronyms are defined in both the Abstract and Conclusions. We also created a list of chemical names in Appendix A, although we defer to the Editor in this regard.

**Some minor editorial points:**

*1. Page 8, line 5 and Table 1: some figures not subscripted*

Subscripted three compounds on L5.

Corrected.

*3. Page 13, lines 27 and 28: the normal convention is to use T for temperatures in kelvin and t for temperatures in degrees Celsius.*

 We have always used *t* for time, but are happy to change to whatever is conventional for ACP in this case, deferring to the Editor.  Since the units are included there is no confusion either way.

*4. Figure 6a: do the concentration plots follow the colour scheme red for arrival, blue for departure? Please make clear in the figure legend.*

No, we used red and blue to more clearly represent two different gases in one panel, but these colours are not related to the pink and blue arrival times.  We added more labels to better explain the top two panels that show two gases each.

[revised manuscript text omitted]